# The Group Robustness is in the Details: Revisiting Finetuning under Spurious Correlations

Tyler LaBonte[1]*    John C. Hill[2]    Xinchen Zhang[2]
Vidya Muthukumar[2,1]    Abhishek Kumar[†]

[1]H. Milton Stewart School of Industrial and Systems Engineering, Georgia Institute of Technology
[2]School of Electrical and Computer Engineering, Georgia Institute of Technology

## Abstract

Modern machine learning models are prone to over-reliance on spurious correlations, which can often lead to poor performance on minority groups. In this paper, we identify surprising and nuanced behavior of finetuned models on worst-group accuracy via comprehensive experiments on four well-established benchmarks across vision and language tasks. We first show that the commonly used class-balancing techniques of mini-batch upsampling and loss upweighting can induce a decrease in worst-group accuracy (WGA) with training epochs, leading to performance no better than without class-balancing. While in some scenarios, removing data to create a class-balanced subset is more effective, we show this depends on group structure and propose a mixture method which can outperform both techniques. Next, we show that scaling pretrained models is generally beneficial for worst-group accuracy, but only in conjunction with appropriate class-balancing. Finally, we identify spectral imbalance in finetuning features as a potential source of group disparities — minority group covariance matrices incur a larger spectral norm than majority groups once conditioned on the classes. Our results show more nuanced interactions of modern finetuned models with group robustness than was previously known. Our code is available at https://github.com/tmlabonte/revisiting-finetuning.

## 1 Introduction

Classification performance in machine learning is sensitive to *spurious correlations*: patterns which are predictive of the target class in the training dataset but not at test time. For example, in computer vision tasks, neural networks are known to utilize the backgrounds of images as proxies for their content [1, 50, 68]. Beyond simple settings, spurious correlations have been identified in high-consequence applications such as criminal justice [8], medicine [70], and facial recognition [33]. In particular, a model's reliance on spurious correlations disproportionately affects its accuracy on *minority groups* which are under-represented in the training dataset; we therefore desire maximizing the model's *group robustness*, quantified by its minimum accuracy on any group [50].

The standard workflow in modern machine learning involves initializing from a pretrained model and finetuning on the downstream dataset using empirical risk minimization (ERM) [62], which minimizes the average training loss. When *group annotations* are available in the training dataset, practitioners utilize a rich literature of techniques to improve worst-group accuracy (WGA) [50, 39, 26]. However, group annotations are often unknown or problematic to obtain (*e.g.*, due to financial, privacy, or fairness concerns). While group robustness methods have been adapted to work without group annotations [31, 72, 47, 29], they remain complex variants on the standard finetuning procedure.

---

*Corresponding author. Email: tlabonte@gatech.edu.
†Work done at Google DeepMind.

38th Conference on Neural Information Processing Systems (NeurIPS 2024).

Hence, it is often unclear to what extent the WGA dynamics of these methods are attributable to details of model finetuning.

In this paper, we take a complementary approach to the methodological literature by pursuing a comprehensive understanding of the *fundamental properties* of model finetuning on four well-established group robustness benchmarks across vision and language tasks. We focus especially on the conjunction of *model scaling* and *class-balancing* — which was recently shown to greatly improve robustness on some datasets [22] — on the worst-group accuracy of the ERM baseline. These considerations enable us to isolate the impact of group disparities on worst-group accuracy, thereby revealing more nuanced behaviors of finetuned models than previously known. In particular, we challenge overarching narratives that "overparameterization helps or hurts distributional robustness" and show striking differences in finetuning performance depending on class-balancing methodology.

In more detail, our main contributions include:

- Identifying two *failure modes* of common class-balancing techniques during fine-tuning: (1) mini-batch upsampling and loss upweighting experience catastrophic collapse with standard hyperparameters on benchmark datasets, and (2) removing data to create a class-balanced subset can harm WGA for certain datasets.
- Proposing a *mixture balancing* method which combines the advantages of two class-balancing techniques and can improve baseline WGA beyond either method.
- Showing that while overparameterization can harm WGA in certain cases, model scaling is generally beneficial for robustness when applied *in conjunction* with appropriate pretraining and class-balancing.
- Identifying a *spectral imbalance* in the top eigenvalues of the group covariances — even when the classes are balanced — and showing that minority group covariance matrices consistently have larger spectral norm conditioned on the classes.

## 1.1 Related work

Here we provide a brief summary of related work along three axes. Throughout the paper, we also provide detailed contextualizations of our results with the most closely related work.

**Spurious correlations.** The proclivity of ERM to rely on spurious correlations has been widely studied [12, 37]. Rectifying this weakness is an important challenge for real-world deployment of machine learning algorithms, as spurious correlations can exacerbate unintended bias against demographic minorities [20, 2, 57, 17, 5] or cause failure in high-consequence applications [33, 8, 70, 42]. Reliance on spurious correlations manifests in image datasets as the usage of visual shortcuts including background [1, 50, 68], texture [11], and secondary objects [48, 52, 54], and in text datasets as the usage of syntactic or statistical heuristics as a substitute for semantic understanding [14, 41, 36].

**Class-balancing and group robustness.** *Group-balancing*, or training with an equal number of samples from each group, has been proposed as a simple yet effective method to improve robustness to spurious correlations [17, 51, 6, 55]. However, group-balancing requires group annotations, which are often unknown or problematic to obtain [31, 72, 47, 29]. On the other hand, *class-balancing*, or training with an equal number of samples from each class, is a well-studied method in long-tailed classification [24, 15, 4]. Recent work has shown that class-balancing is a surprisingly powerful method for improving worst-group accuracy which does not require group annotations [22, 29, 7, 53]. In particular, [22] study the WGA dynamics of two common class-balancing methods: removing data from the larger classes (which we call *subsetting*) and upsampling the smaller classes (which we call *upsampling*). Our results complement those of [22] and show more nuanced effects of class-balancing than previously known; we provide additional contextualization with [22] in Section 3.1. We show similar nuanced behavior of *upweighting* smaller classes in the loss function, a popular method in the group-balancing setting [31, 47, 55] which [22] did not study.

**Overparameterization and distributional robustness.** While the accepted empirical wisdom is that overparameterization improves *in-distribution* test accuracy [40, 71], the relationship between overparameterization and robustness is incompletely understood. [51] considered a class of ResNet-18 architectures and showed that increasing model *width* reduces worst-group accuracy on the

Waterbirds and CelebA datasets when trained with class-imbalanced ERM — this contrasts with the improvement in average accuracy widely observed in practice (see, *e.g.*, [38]). Conversely, [19] showed a benefit of overparameterization in robustness to "natural" covariate shifts, which are quite different from spurious correlations [27]. On the mathematical front, [59, 35] showed that overparameterization in random feature models trained to completion improves robustness to a wide class of covariate shifts.However, both the optimization trajectory and statistical properties of random features are very different from neural networks (see, *e.g.*, [13]). Closely related to our work, [46] investigated pretrained ResNet, VGG, and BERT models, and showed that overparameterization does not harm WGA. Our results complement those of [46] with a richer setup and show that class-balancing — which they do not study — can greatly impact model scaling behavior.

## 2   Preliminaries

**Setting.**   We consider classification tasks with input domain $\mathbb{R}^n$ and target classes $\mathcal{Y} \subset \mathbb{N}$. Suppose $\mathcal{S}$ is a set of *spurious features* such that each example $x \in \mathbb{R}^n$ is associated with exactly one feature $s(x) \in \mathcal{S}$. The dataset is then partitioned into *groups* $\mathcal{G}$, defined by the Cartesian product of classes and spurious features $\mathcal{G} = \mathcal{Y} \times \mathcal{S}$. Given a dataset of $m$ training examples, we define the set of indices of examples which belong to some group $g \in \mathcal{G}$ or class $y \in \mathcal{Y}$ by $\Omega_g \subseteq \{1, \dots, m\}$ and $\Omega_y \subseteq \{1, \dots, m\}$, respectively. Then, the *majority group(s)* is defined by the group(s) that maximize $|\Omega_g|$. All other groups are designated as *minority groups*. Further, the *worst group(s)*[3] is defined by the group(s) which incur minimal test accuracy. We define majority and minority classes similarly. Because groups are defined by the Cartesian product of classes and spurious features, all training examples in a particular group are identically labeled, and therefore *a group is a subset of a class*.

We desire a model which, despite group imbalance in the training dataset, enjoys roughly uniform performance over $\mathcal{G}$. Therefore, we evaluate *worst-group accuracy* (WGA), *i.e.*, the minimum accuracy among all groups [50]. We will also be interested in the relative performance on groups *within the same class*, and we thereby define the *majority group within a class* $y \in \mathcal{Y}$ as the group which maximizes $|\Omega_g|$ over all $g \in \{g \in \mathcal{G} : y \in g\}$. Other groups are designated as the minority groups within that class. For example, referring to the Waterbirds section of Table 2, groups 1 and 2 are the minority groups within classes 0 and 1, respectively.

**Class-balancing.**   A dataset is considered to be *class-balanced* if it is composed of an equal number of training examples from each class in expectation over the sampling probabilities. We compare three class-balancing techniques: *subsetting*, *upsampling*, and *upweighting*. We describe each below:

- In *subsetting*, every class is set to the same size as the smallest class by removing the appropriate amount of data from each larger class uniformly at random. This procedure is performed only once, and the subset is fixed prior to training.

- In *upsampling*, the entire dataset is utilized for training with a typical stochastic optimization algorithm, but the sampling probabilities of each class are adjusted so that mini-batches are class-balanced in expectation. To draw a single example, we first sample $y \sim \text{Unif}(\mathcal{Y})$, then sample $x \sim \hat{p}(\cdot \mid y)$ where $\hat{p}$ is the *empirical* distribution on training examples.

- In *upweighting*, the minority class samples are directly upweighted in the loss function according to the ratio of majority class data to minority class data, called the *class-imbalance ratio*. Specifically, if the loss function is $\ell(f(x), y)$ for model $f$, example $x$, and class label $y$, the upweighted loss function is $\gamma \ell(f(x), y)$ where $\gamma$ is defined as the class-imbalance ratio for minority class data and $1$ for majority class data. It is worth noting that upweighting is equivalent to upsampling in expectation over the sampling probabilities.

Note that the terminology for these class-balancing techniques is not consistent across the literature. For example, [22] call subsetting *subsampling* (denoted SUBY) and upsampling *reweighting* (denoted RWY). On the other hand, [55] call (group-wise) subsetting *downsampling* and use *upweighting* to describe increasing the weight of minority group samples in the loss function.

---

[3]Note that, as is standard in the empirical literature on distributional robustness, majority, minority and worst groups are defined with respect to the empirical training distribution, as this is all that we have access to. Moreover, test accuracy is typically maximized by the majority group and minimized by a minority group, though this is not always the case.

**Datasets and models.** We study four classification datasets, two in the vision domain and two in the language domain, which are well-established as benchmarks for group robustness. We summarize each dataset below and provide additional numerical details in Appendix A.1.

- *Waterbirds* [64, 63, 50] is an image dataset wherein birds are classified as land species ("landbirds") or water species ("waterbirds"). The spurious feature is the image background: more landbirds are present on land backgrounds and vice versa.[4]

- *CelebA* [33, 50] is an image dataset classifying celebrities as blond or non-blond. The spurious feature is gender, with more blond women than blond men in the training dataset.

- *CivilComments* [3, 27] is a language dataset wherein online comments are classified as toxic or non-toxic. The spurious feature is the presence of one of the following categories: male, female, LGBT, black, white, Christian, Muslim, or other religion.[5] More toxic comments contain one of these categories than non-toxic comments, and vice versa.

- *MultiNLI* [65, 50] is a language dataset wherein pairs of sentences are classified as a contradiction, entailment, or neither. The spurious feature is a negation in the second sentence — more contradictions have this property than entailments or neutral pairs.

Waterbirds is class-imbalanced with a majority/minority class ratio of 3.31:1, CelebA a ratio of 5.71:1, and CivilComments a ratio of 7.85:1. MultiNLI is class-balanced *a priori*. Since the Waterbirds dataset has a shift in group proportion from train to test, we weight the group accuracies by their proportions in the training set when reporting the test average accuracy [50].

We utilize ResNet [18], ConvNeXt-V2 [67], and Swin Transformer [32] models pretrained on ImageNet-1K [49] for Waterbirds and CelebA, and a BERT [9] model pretrained on Book Corpus [73] and English Wikipedia for CivilComments and MultiNLI. We use the AdamW optimizer [34] for finetuning on three independent seeds, randomizing both mini-batch order and any other stochastic procedure such as subsetting, and we report error bars corresponding to one standard deviation. We do not utilize early-stopping: instead, to consider the impact of overparameterization in a holistic way, we train models to completion to properly measure the overfitting effect.[6] This can result in longer training than commonly seen in the literature (*e.g.*, we finetune on CelebA for about $3\times$ more gradient steps than is standard). See Appendix A.2 for further training details.

## 3 Nuanced effects of class-balancing on group robustness

We now present our first set of results, which shows that the choice of class balancing method greatly impacts the group robustness of the ERM baseline.

### 3.1 Catastrophic collapse of class-balanced upsampling and upweighting

In a recent paper, [29] observed that contrary to the central hypothesis underlying the Just Train Twice method [31], the worst-group accuracy of ERM decreases dramatically with training epochs on CelebA and CivilComments; however, they provide no explanation for this phenomenon. In this section, we show that this degradation of WGA is due to their choice of class-balancing method (*i.e.*, upsampling). Specifically, ERM finetuned with upsampling experiences a *catastrophic collapse* in test WGA over the course of training, a phenomenon that was previously only noticed in synthetic datasets with a linear classifier [22]. Moreover, while [22] state that class-balanced subsetting is not recommended in practice, we show that it can in fact improve WGA conditional on the lack of of a small minority group *within the majority class*. Finally, we show that class-balanced upweighting — a popular technique which [22] do not study — experiences a similar WGA collapse as upsampling.

We finetune a ConvNeXt-V2 Base on Waterbirds and CelebA and a BERT Base on CivilComments, and we compare the subsetting, upsampling, and upweighting techniques to a class-imbalanced

---

[4]We note that the Waterbirds dataset is known to contain incorrect labels [56]. We report results on the original, un-corrected version as is standard in the literature.

[5]This version of CivilComments has four groups, used in this work and by [50, 22, 23, 26, 29]. There is another version where the identity categories are not collapsed into one spurious feature; that version is used by [31, 72, 47]. Both versions use the WILDS split [27].

[6]To be more specific, we finetune ConvNeXt-V2 Base roughly to a training loss of $10^{-4}$ on Waterbirds and $10^{-3}$ on CelebA, and BERT Base roughly to a training loss of $10^{-3}$ on CivilComments and $10^{-2}$ on MultiNLI.

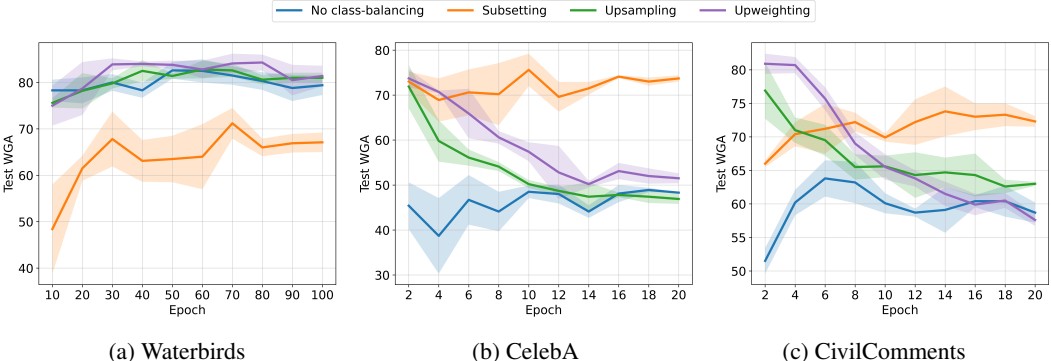

| No class-balancing | Subsetting | Upsampling | Upweighting |

(a) Waterbirds            (b) CelebA            (c) CivilComments

Figure 1: **Class-balanced upsampling and upweighting experience catastrophic collapse.** We compare *subsetting*, wherein data is removed to set every class to the same size as the smallest class, *upsampling*, wherein the sampling probabilities of each class are adjusted so that the mini-batches are class-balanced in expectation, and *upweighting*, wherein the loss for the smaller classes is scaled by the class-imbalance ratio. We observe a catastrophic collapse over the course of training of upsampling and upweighting on CelebA and CivilComments, the more class-imbalanced datasets. Subsetting reduces WGA on Waterbirds because it removes data from the small minority group within the majority class. MultiNLI is class-balanced *a priori*, so we do not include it here.

baseline. Our results are displayed in Figure 1, with additional models in Appendix B. On CelebA and CivilComments, the more class-imbalanced datasets, upsampling and upweighting both experience catastrophic collapse over the course of training. We believe this collapse is caused by overfitting to the minority group within the minority class; any individual point from this group is sampled far more often during upsampling and weighted far more heavily during upweighting, causing overfitting during long training runs. In fact, upsampling does even worse on CelebA than observed in [29] because we train $3\times$ longer to ensure convergence. With that said, optimally tuned early-stopping appears to mitigate the collapse (as previously noticed by [22] in a toy setting).

Our experiments also highlight a previously unnoticed disadvantage of class-balanced subsetting: if there is a small minority group in the majority class, subsetting will further reduce its proportion and harm WGA. For example, in the Waterbirds dataset, the species (landbirds/waterbirds) is the class label and the background (land/water) is the spurious feature; *landbirds/water* is a small minority group within the majority class (landbirds). When landbirds is cut by $3.31\times$, the landbirds/water group greatly suffers, harming WGA. On the other hand, in the CelebA dataset, the hair color (non-blond/blond) is the class label and the gender (female/male) is the spurious feature; the only small minority group is *blond/male*, while the groups are nearly balanced in the majority class. In this case, subsetting preserves blond/male examples and increases their proportion, helping WGA.

Finally, while upsampling and upweighting have similar WGA dynamics – perhaps as expected, as they are equivalent in expectation over the sampling mechanism — both differ greatly from subsetting. Recently, [55] proved a theoretical equivalence between subsetting and upsampling of the *groups* in the *population* setting, *i.e.*, assuming access to the training distribution. The equivalence of upsampling and upweighting would then imply that all three objectives are optimized by the same solution. However, our results suggest this may not hold in the real-world *empirical* setting, where subsetting has distinctly different behavior, and model parameters may outnumber training examples. As previously mentioned, this may be due to overfitting to minority class data repeated often during training; theoretically investigating this discrepancy is an important future direction.

**Contextualization with previous work.** Our observations explain the decrease in WGA of CelebA and CivilComments noticed by [29], a phenomenon which they left unresolved. Our result implies that group robustness methods which assume that WGA increases during training, such as Just Train Twice [31], may only be justified with appropriate class-balancing. [22] show that upsampling can cause catastrophic collapse in WGA, but only in a synthetic dataset with a linear classifier. In realistic datasets, [22] perform extensive hyperparameter tuning (using group labels, which may be unrealistic) to achieve good results with upsampling, while we show that catastrophic collapse can occur in the same datasets when standard hyperparameters are used. Moreover, [22] state that class-balanced

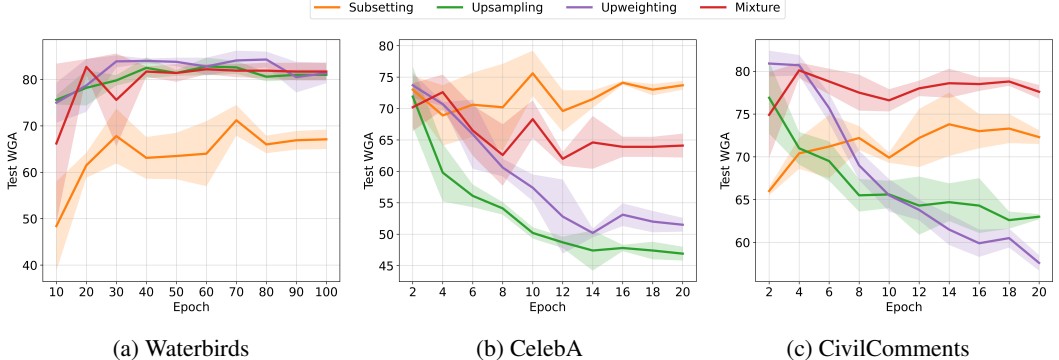

(a) Waterbirds       (b) CelebA       (c) CivilComments

Figure 2: **Mixture balancing mitigates catastrophic collapse of upsampling and upweighting.** We propose a class-balanced *mixture method*, which combines subsetting and upsampling by first drawing a class-imbalanced subset uniformly at random from the dataset, then adjusting sampling probabilities so that mini-batches are balanced in expectation. Our method increases exposure to majority class data without over-sampling the minority class. Remarkably, mixture balancing outperforms all three class-balancing methods on Waterbirds and CivilComments, and while it does not outperform subsetting on CelebA, it significantly alleviates the WGA collapse experienced by upsampling and upweighting. MultiNLI is class-balanced *a priori*, so we do not include it here.

subsetting is not recommended in practice, but we show that subsetting can be effective except when there is a small minority group within the majority class, a previously unnoticed nuance. Finally, we show that subsetting experiences different WGA dynamics from upsampling and upweighting in the empirical setting, suggesting additional complexity compared to the population setting results of [55].

> Without extensive tuning, class-balanced *upsampling* and *upweighting* can induce WGA no better than without class-balancing. While class-balanced *subsetting* can improve WGA, practitioners should use caution if a small minority group is present within the majority class.

### 3.2 Mixture balancing: interpolating between subsetting and upsampling

To mitigate the catastrophic collapse of class-balanced upsampling and upweighting, we propose a simple *mixture method* which interpolates between subsetting and upsampling. Our method increases exposure to majority class data without over-sampling the minority class, which can improve WGA and mitigate overfitting to the minority group. We first create a data subset with a specified *class-imbalance ratio* by removing data from the larger classes uniformly at random until the desired (smaller) ratio is achieved. Next, we perform ERM finetuning on this subset by adjusting sampling probabilities so that mini-batches are balanced in expectation. Using a class-imbalance ratio of 1:1 reduces to subsetting, and using the original class-imbalance ratio reduces to upsampling.

We finetune ConvNeXt-V2 Base on Waterbirds and CelebA and BERT Base on CivilComments, and we compare our class-balanced mixture method to the subsetting, upsampling, and upweighting techniques. The results of our experiments are displayed in Figure 2. We plot the performance of our mixture method with the best class-imbalance ratio during validation; an ablation study varying the ratio is included in Appendix B. Remarkably, mixture balancing outperforms all three class-balancing methods on Waterbirds and CivilComments, and while it does not outperform subsetting on CelebA, it significantly alleviates the WGA collapse experienced by upsampling.

Next, we perform an ablation of the necessity of subsetting in mixture balancing. We compare our method with an implementation which eschews subsetting, instead adjusting sampling probabilities so that the mini-batches have a particular class ratio in expectation. For example, instead of performing upsampling on a 2:1 class-imbalanced subset, we upsample the majority class by a ratio of 2:1 on the entire dataset. The results of our ablation are included in Appendix B; our mixture method outperforms the alternative, which incompletely corrects for class imbalance.

Table 1: **Mixture balancing is robust to model selection without group annotations.** We compare the best class-balancing method during validation with and without group annotations. Both worst-class accuracy [69] and the bias-unsupervised validation score of [60] are effective for model selection without group annotations, often choosing the same method or mixture ratio as worst-group accuracy (WGA) validation. We list the method maximizing each metric and its average WGA over 3 seeds.

| Validation Metric | Group Anns | Waterbirds | CelebA | CivilComments |
|---|---|---|---|---|
| Bias-unsupervised Score | ✗ | Upsampling (79.9) | Subsetting (74.1) | Mixture 3:1 (77.6) |
| Worst-class Accuracy | ✗ | Mixture 2:1 (81.1) | Subsetting (74.1) | Mixture 3:1 (77.6) |
| Worst-group Accuracy | ✓ | Mixture 2:1 (81.1) | Subsetting (74.1) | Mixture 3:1 (77.6) |

**Note on validation.**    In Figure 2, we plot the best class-imbalance ratio achieved using validation on a group annotated held-out set. While this is a common assumption in the literature [50, 31, 23, 26], it is nevertheless unrealistic when the training set does not have any group annotations. Therefore, we compare with both worst-class accuracy [69] and the *bias-unsupervised validation score* of [60], which do not use any group annotations for model selection. In Table 1 we list the method which maximizes each validation metric as well as its average WGA. Overall, we show both methods are effective for model selection, often choosing the same method or mixture ratio as WGA validation.

**Contextualization with previous work.**    Increasing exposure to majority class data without over-sampling the minority class was previously explored by [26], who proposed averaging the weights of logistic regression models trained on ten independent class-balanced subsets. However, this method only works for *linear* models — as nonlinear models cannot be naively averaged — and requires multiple training runs, which is computationally infeasible for neural networks. In comparison, our mixture method is a simple and efficient alternative which extends easily to nonlinear models.

> The catastrophic collapse of class-balanced upsampling and upweighting can be mitigated by a *mixture method*. It increases exposure to majority class data without over-sampling the minority class and can improve baseline WGA beyond either technique.

## 4    Model scaling improves WGA of class-balanced finetuning

The relationship between overparameterization and group robustness has been well-studied, with often conflicting conclusions [51, 59]. In this section, we study the impact of model scaling on worst-group accuracy in a new setting — finetuning pretrained models — which more closely resembles practical use-cases. Importantly, we evaluate the impact of model scaling *in conjunction with class-balancing* to isolate the impact of group inequities on WGA as a function of model size. We find that with appropriate class-balancing, overparameterization can in fact significantly improve WGA over a very wide range of parameter scales, including before and after the interpolation threshold. On the other hand, scaling on imbalanced datasets or with the wrong balancing technique can harm robustness.

We take advantage of advancements in efficient architectures [61, 67] to finetune pretrained models in a wide range of scales from 3.4M to 101M parameters. We study six different sizes of ImageNet1K-pretrained ConvNeXt-V2 and five different sizes of Book Corpus/English Wikipedia pretrained BERT; specifications for each model size are included in Appendix A.2. Our results are displayed in Figure 3, and we include results for Swin Transformer in Appendix C.

We find that model scaling is beneficial for group robustness in conjunction with appropriate class-balancing, with improvements of up to 12% WGA for interpolating models and 40% WGA for non-interpolating models. This comes in stark contrast to scaling on class-imbalanced datasets or with the wrong class-balancing technique, which shows either a neutral trend or decrease in WGA — the most severe examples being on CivilComments. With respect to interpolating models, CivilComments WGA decreases slightly after the interpolation threshold, while Waterbirds and CelebA continue to improve well beyond interpolation; on the other hand, BERT never interpolates MultiNLI, greatly increasing robustness at scale. It is unclear why Waterbirds and CelebA experience

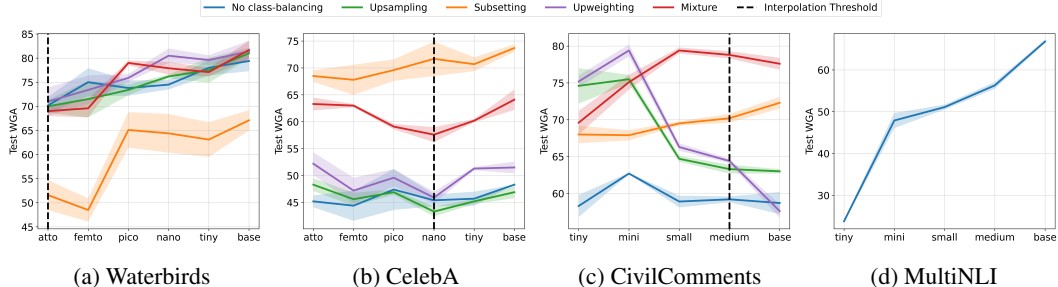

| (a) Waterbirds | (b) CelebA | (c) CivilComments | (d) MultiNLI |

Figure 3: **Scaling class-balanced pretrained models can improve worst-group accuracy.** We finetune each model size starting from pretrained checkpoints and plot the test worst-group accuracy (WGA) as well as the interpolation threshold, where model reaches $100\%$ training accuracy. We find model scaling is generally beneficial for WGA *only in conjunction* with appropriate class-balancing, and scaling on imbalanced datasets or with the wrong method can harm robustness. Note MultiNLI is class-balanced *a priori* and is not interpolated. See Appendix C for training accuracy plots.

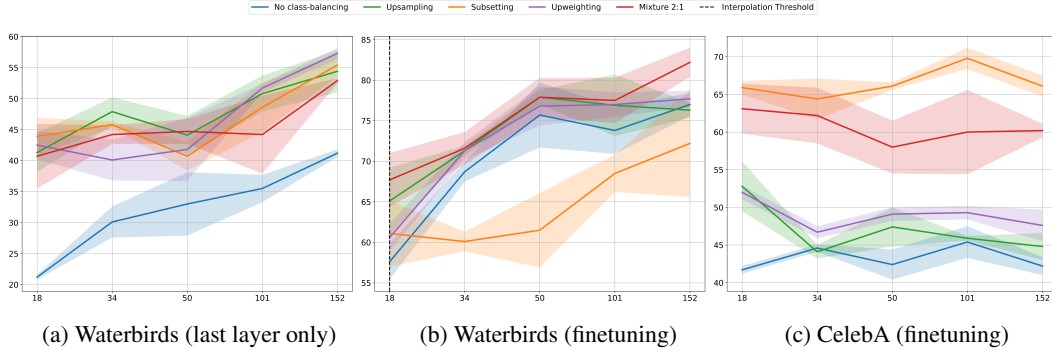

| (a) Waterbirds (last layer only) | (b) Waterbirds (finetuning) | (c) CelebA (finetuning) |

Figure 4: **Class-balancing greatly affects ResNet scaling results of [46].** We contrast the ResNet scaling behavior of [46] — who do not use class-balancing — to the scaling of class-balanced ResNets. We finetune each model size starting from pretrained checkpoints and plot the test worst-group accuracy (WGA), as well as the interpolation threshold, where the model reaches $100\%$ training accuracy. On Waterbirds, we find that class-balancing enables a much more beneficial trend during model scaling. On CelebA, class-balancing greatly increases baseline WGA but does not affect scaling behavior (in contrast to the ConvNeXt-V2 plots in Figure 3). We use SGD for last-layer training and AdamW for full finetuning. See Appendix C for training accuracy plots.

different behavior from CivilComments interpolation — the toy linear model of [51] suggests a benign "spurious-core information ratio", but a complete understanding is left to future investigation.

The most closely related work to ours is [46], who study the impact of scaling pretrained ResNet models on group robustness. However, because their experiments do not employ any form of class-balancing, their conclusions may be overly pessimistic. We replicate their experiments with our hyperparameters and contrast with our results using class-balancing in Figure 4. We find that class-balancing greatly affects their results: on Waterbirds, class-balancing enables a much more beneficial trend during model scaling regardless of whether a linear probe or the entire model is trained. Moreover, while class-balancing increases baseline WGA on CelebA but does not affect scaling behavior, we observe a more positive WGA trend when scaling ConvNeXt-V2 in Figure 3.

**Contextualization with previous work.** While previous work has primarily studied either linear probing of pretrained weights or training small models from scratch [51, 59], we study full finetuning of large-scale pretrained models and show that class-balancing can have a major impact on scaling behavior. We compare directly with the most closely related work, [46], and show that class-balancing can either induce strikingly different scaling behavior or greatly increase baseline WGA. Overall, training with class-balancing allows us to isolate the impact of group inequities on robustness and more precisely observe the often-beneficial trend of model scaling for worst-group accuracy.

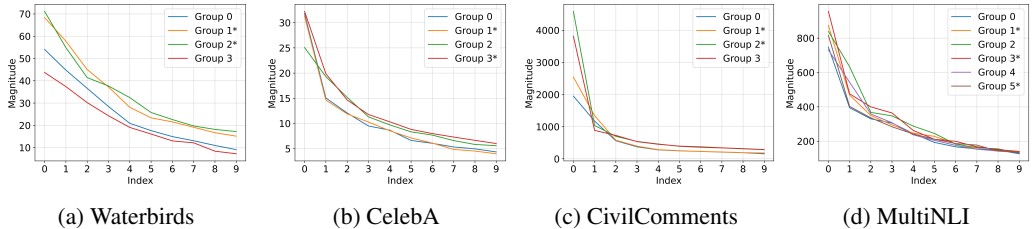

| (a) Waterbirds | (b) CelebA | (c) CivilComments | (d) MultiNLI |

Figure 5: **Group disparities are visible in the top eigenvalues of the group covariance matrices.** We visualize the mean, across 3 experimental trials, of the top 10 eigenvalues of the group covariance matrices for a ConvNeXt-V2 Nano finetuned on Waterbirds and CelebA and a BERT Small finetuned on CivilComments and MultiNLI. The standard deviations are omitted for clarity. The models are finetuned using the best class-balancing method from Section 3 for each dataset. The group numbers are detailed in Table 2 and the minority groups within each class are denoted with an asterisk. The largest $\lambda_1$ in each case belongs to a minority group, though it may not be the *worst* group, and minority group eigenvalues are overall larger than majority group eigenvalues within the same class.

> While overparameterization can sometimes harm WGA, pretraining and appropriate class-balancing make scaling generally beneficial. Moreover, modern language datasets are complex enough that standard models *do not interpolate*, greatly improving robustness at scale.

## 5 Spectral imbalance may exacerbate group disparities

In a recent paper, [25] propose *spectral imbalance* of class covariance matrices, or differences in their eigenspectrum, as a source of disparities in accuracy *across classes* even when balanced. Here, we examine whether similar insights hold in the group robustness setting. Our observations reveal surprising nuances in the behavior of *group-wise spectral imbalance*; nevertheless, we conclude that spectral imbalance may play a similar role in modulating WGA after class-balancing is applied.

Let us denote by $z_i$ the feature vector corresponding to a sample $x_i$ (*i.e.,* the vectorized output of the *penultimate* layer). Recall from Section 2 that $\Omega_g$ is the set of indices of samples which belong to group $g$. We further define $\bar{z}_g$ to be the empirical mean of features with group $g$. To obtain the estimated eigenspectrum, we first compute the empirical covariance matrix for group $g \in \mathcal{G}$ by

$$\boldsymbol{\Sigma}_g = \frac{1}{|\Omega_g|} \sum_{i \in \Omega_g} (z_i - \bar{z}_g)(z_i - \bar{z}_g)^\top.$$

We then compute the eigenvalue decomposition $\boldsymbol{\Sigma}_g = \mathbf{V}_g \boldsymbol{\Lambda}_g \mathbf{V}_g^{-1}$, where $\boldsymbol{\Lambda}_g$ is a diagonal matrix with non-negative entries $\lambda_i^{(g)}$ and the columns of $\mathbf{V}_g$ are the eigenvectors of $\boldsymbol{\Sigma}_g$. Without loss of generality, we assume $\lambda_1^{(g)} \geq \lambda_2^{(g)} \geq \cdots \geq \lambda_m^{(g)}$ where $m$ is the rank of $\boldsymbol{\Sigma}_g$.

We compute the group covariance matrices using a ConvNeXt-V2 Nano model for Waterbirds and CelebA, and a BERT Small model for CivilComments and MultiNLI. We plot the top 10 eigenvalues of each group covariance matrix in Figure 5. Even though we finetune with class-balancing, disparities in eigenvalues across groups are clearly visualized in Figure 5, especially for the largest eigenvalues. We include extensions to the top 50 eigenvalues and class covariance matrices in Appendix D.

Close observation of Figure 5 yields interesting findings. First, the group $g^*$ that maximizes $\lambda_1^{(g)}$ in each case belongs to a minority group; though, importantly, it may not belong to the *worst* group. This is different from the findings of [25], who showed that the largest eigenvalues typically belong to the *worst-performing* class. Second, we find that minority group eigenvalues are overall larger than majority group eigenvalues, but only when *conditioned on the class*. A majority group belonging to one class may have larger eigenvalues than a minority group belonging to another class, but there exists a consistent spectral imbalance between majority and minority groups within the same class.[7]

---

[7]For example, in Figure 5c, the spectrum for group 3 (the majority group within class 1) is larger than the spectrum for group 1 (the minority group within class 0). However, conditioning on the class, we find that the

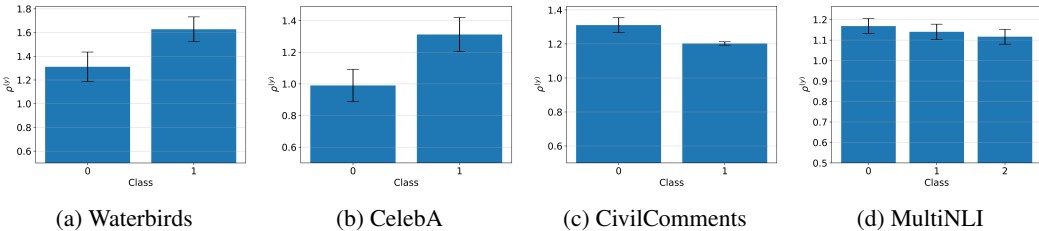

| (a) Waterbirds | (b) CelebA | (c) CivilComments | (d) MultiNLI |

Figure 6: **Group-wise spectral imbalance is apparent once conditioned on the classes.** We plot the mean and standard deviation, across 3 experimental trials, of the intra-class spectral norm ratio $\rho^{(y)}$, or the ratio of the top eigenvalues of the minority and majority group covariance matrices, for each class $y \in \mathcal{Y}$. We compute this metric using a finetuned ConvNeXt-V2 Nano on Waterbirds and CelebA and a finetuned BERT Small on CivilComments and MultiNLI, each using the best class-balancing method from Section 3 for each dataset. The key observation is that $\rho^{(y)}$ is at least one for all classes $y \in \mathcal{Y}$ (except a single seed for class 0 on CelebA), illustrating a group disparity captured by the eigenspectrum once we condition on the classes.

To quantify this group-wise spectral imbalance, we introduce a new metric called the *intra-class spectral norm ratio*. Suppose $g_{\min}(y)$ and $g_{\mathrm{maj}}(y)$ are the minority and majority groups within a particular class $y \in \mathcal{Y}$. Then, we define the intra-class spectral norm ratio by $\rho^{(y)} := \lambda_1^{(g_{\min}(y))}/\lambda_1^{(g_{\mathrm{maj}}(y))}$. While $\rho^{(y)}$ only considers the top eigenvalue and not the entire spectrum, the *absolute* magnitude of individual eigenvalues was found in [25] to correlate best with worst-class accuracy. We note that $\rho^{(y)}$ considers only the top eigenvalue and not the entire spectrum, since the magnitude of the top eigenvalues was found in [25] to correlate best with worst-class accuracy. We plot the intra-class spectral norm ratios for each dataset in Figure 6; notably, they are always at least one (except for a single seed on CelebA), showing the group disparity captured by the eigenspectrum.

Finally, in Table 5 (deferred to Appendix D), we compare the class with the largest $\rho^{(y)}$ to the class with the largest disparity in group test accuracies, *i.e.*, $\mathrm{Acc}(g_{\mathrm{maj}}(y)) - \mathrm{Acc}(g_{\min}(y))$. We see that in most cases these classes correspond, suggesting an *explanatory power* of the intra-class spectral norm ratio. In particular, this correspondence is consistent throughout all trials of CelebA and CivilComments, the most class-imbalanced datasets we study.

**Contextualization with previous work.** Our spectral analysis of the group covariance matrices is inspired by [25]. We both study class-balanced settings, with the key difference that they study *class* disparities instead of group disparities. However, we show a more nuanced impact of spectral imbalance across both classes and groups, *i.e.*, spectral imbalance is more prevalent *between majority and minority groups within to the same class*, rather than across groups globally.

> Spectral imbalance in the group covariance matrices may exacerbate group disparities even when the classes are balanced. While the worst-group covariance may not have largest spectral norm, the minority group spectra are consistently larger *conditioned on the class*.

## 6 Discussion

In this paper, we identified nuanced impacts of class-balancing and model scaling on worst-group accuracy, as well as a spectral imbalance in the group covariance matrices. Overall, our work calls for a more thorough investigation of generalization in the presence of spurious correlations to unify the sometimes contradictory perspectives in the literature. We hope that, as the community continues to develop group robustness methods with increasing performance and complexity, researchers and practitioners alike remain cognizant of the disproportionate impact of the details.

---

spectrum for group 2 (the minority group within class 1) is larger than that of group 3, and the spectrum of group 1 is larger than that of group 0 (the majority group within class 0).

**Acknowledgments.** We thank Google Cloud for the gift of compute credits, Jacob Abernethy for additional compute assistance, and Chiraag Kaushik for helpful discussions. T.L. acknowledges support from the DoD NDSEG Fellowship. V.M. acknowledges support from the NSF (awards CCF-223915 and IIS-2212182), Google Research, Adobe Research and Amazon Research.

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

# A  Additional Details for Section 2

## A.1  Dataset Composition

Table 2: **Dataset composition.** We study four well-established benchmarks for group robustness across vision and language tasks. The class probabilities change dramatically when conditioned on the spurious feature. Note that Waterbirds is the only dataset that has a distribution shift and MultiNLI is the only dataset which is class-balanced *a priori*. The minority groups within each class are denoted by an asterisk in the "Num" column. Probabilities may not sum to 1 due to rounding.

| Dataset | Group $g$ | | | Training distribution $\hat{p}$ | | | Data quantity | | |
|---|---|---|---|---|---|---|---|---|---|
| | Num | Class $y$ | Spurious $s$ | $\hat{p}(y)$ | $\hat{p}(g)$ | $\hat{p}(y\mid s)$ | Train | Val | Test |
| Waterbirds | 0 | landbird | land | .768 | .730 | .984 | 3498 | 467 | 2225 |
| | 1* | landbird | water | | .038 | .148 | 184 | 466 | 2225 |
| | 2* | waterbird | land | .232 | .012 | .016 | 56 | 133 | 642 |
| | 3 | waterbird | water | | .220 | .852 | 1057 | 133 | 642 |
| CelebA | 0 | non-blond | female | .851 | .440 | .758 | 71629 | 8535 | 9767 |
| | 1* | non-blond | male | | .411 | .980 | 66874 | 8276 | 7535 |
| | 2 | blond | female | .149 | .141 | .242 | 22880 | 2874 | 2480 |
| | 3* | blond | male | | .009 | .020 | 1387 | 182 | 180 |
| CivilComments | 0 | neutral | no identity | .887 | .551 | .921 | 148186 | 25159 | 74780 |
| | 1* | neutral | identity | | .336 | .836 | 90337 | 14966 | 43778 |
| | 2* | toxic | no identity | .113 | .047 | .079 | 12731 | 2111 | 6455 |
| | 3 | toxic | identity | | .066 | .164 | 17784 | 2944 | 8769 |
| MultiNLI | 0 | contradiction | no negation | .333 | .279 | .300 | 57498 | 22814 | 34597 |
| | 1* | contradiction | negation | | .054 | .761 | 11158 | 4634 | 6655 |
| | 2 | entailment | no negation | .334 | .327 | .352 | 67376 | 26949 | 40496 |
| | 3* | entailment | negation | | .007 | .104 | 1521 | 613 | 886 |
| | 4 | neither | no negation | .333 | .323 | .348 | 66630 | 26655 | 39930 |
| | 5* | neither | negation | | .010 | .136 | 1992 | 797 | 1148 |

## A.2  Training details

We utilize ResNet [18], ConvNeXt-V2 [67], and Swin Transformer [32] models pretrained on ImageNet-1K [49] for Waterbirds and CelebA, and a BERT [9] model pretrained on Book Corpus [73] and English Wikipedia for CivilComments and MultiNLI. These pretrained models are used as the initialization for ERM finetuning under the cross-entropy loss. We use standard ImageNet normalization with standard flip and crop data augmentation for the vision tasks and BERT tokenization for the language tasks [23]. Our implementation uses the following packages: NumPy [16], PyTorch [44, 45], Lightning [10], TorchVision [58], Matplotlib [21], Transformers [66], and Milkshake [28].

To our knowledge, the licenses of Waterbirds and CelebA are unknown. CivilComments is released under the CC0 license, and information about MultiNLI's license may be found in [65].

Our experiments were conducted on four Google Cloud Platform (GCP) 16GB Nvidia Tesla P100 GPUs and two local 24GB Nvidia RTX A5000 GPUs. The spectral imbalance experiments in Section 5 were conducted on a GCP system with a 16-core CPU and 128GB of RAM. We believe our work could be reproduced for under $5000 in GCP compute credits, with a majority of that compute going towards running experiments over multiple random seeds.

We list model scaling parameters in Table 3 and hyperparameters used for each dataset in Table 4. ConvNeXt-V2, ResNet and Swin Transformers are composed of four separate "stages", and we list the depths of these stages individually in Table 3. All of these configurations are standard in the literature. The smaller BERT models were introduced by [61]. We perform model selection only for our mixture balancing method (see Table 1) and not for the ERM finetuning hyperparameters, most of which are standard in the literature [50, 22, 23]. For the last-layer training experiments in Figure 4 and Figure 11, we use SGD with learning rate $10^{-3}$ and train for 20 epochs. Different from previous work, we train CelebA for about $3\times$ more gradient steps than usual to ensure convergence, and we

double the batch size for CivilComments and MultiNLI to increase training stability (we also double the epochs to hold the number of gradient steps constant).

Table 3: Model scaling parameters.

(a) **ConvNeXt-V2 parameters.**

| Size | Width | Depth (4 stages) | Params |
|------|-------|------------------|--------|
| Atto | 40 | $(2, 2, 6, 2)$ | 3.4M |
| Femto | 48 | $(2, 2, 6, 2)$ | 4.8M |
| Pico | 64 | $(2, 2, 6, 2)$ | 8.6M |
| Nano | 80 | $(2, 2, 8, 2)$ | 15.0M |
| Tiny | 96 | $(3, 3, 9, 3)$ | 27.9M |
| Base | 128 | $(3, 3, 27, 3)$ | 87.7M |

(b) **BERT parameters.**

| Size | Width | Depth | Params |
|------|-------|-------|--------|
| Tiny | 2 | 128 | 4.4M |
| Mini | 4 | 256 | 11.2M |
| Small | 4 | 512 | 28.8M |
| Medium | 8 | 512 | 41.4M |
| Base | 12 | 768 | 109M |

(c) **ResNet parameters.**

| Size | Width (4 stages) | Depth (4 stages) | Params |
|------|------------------|------------------|--------|
| 18 | $(64, 128, 256, 512)$ | $(2, 2, 2, 2)$ | 11.2M |
| 34 | $(64, 128, 256, 512)$ | $(3, 4, 6, 3)$ | 21.3M |
| 50 | $(256, 512, 1024, 2048)$ | $(3, 4, 6, 3)$ | 23.5M |
| 101 | $(256, 512, 1024, 2048)$ | $(3, 4, 23, 3)$ | 42.5M |
| 152 | $(256, 512, 1024, 2048)$ | $(3, 8, 36, 3)$ | 58.1M |

(d) **Swin Transformer parameters.**

| Size | Width | Depth (4 stages) | Params |
|------|-------|------------------|--------|
| Tiny | 96 | $(2, 2, 6, 2)$ | 29M |
| Small | 96 | $(2, 2, 18, 2)$ | 50M |
| Base | 128 | $(2, 2, 18, 2)$ | 88M |

Table 4: ERM finetuning hyperparameters.

| Dataset | Optimizer | Initial LR | LR schedule | Batch size | Weight decay | Epochs |
|---------|-----------|------------|-------------|------------|--------------|--------|
| Waterbirds | AdamW | $1 \times 10^{-5}$ | Cosine | 32 | $1 \times 10^{-4}$ | 100 |
| CelebA | AdamW | $1 \times 10^{-5}$ | Cosine | 32 | $1 \times 10^{-4}$ | 20 |
| CivilComments | AdamW | $1 \times 10^{-5}$ | Linear | 32 | $1 \times 10^{-4}$ | 20 |
| MultiNLI | AdamW | $1 \times 10^{-5}$ | Linear | 32 | $1 \times 10^{-4}$ | 20 |

# B   Additional Experiments for Section 3

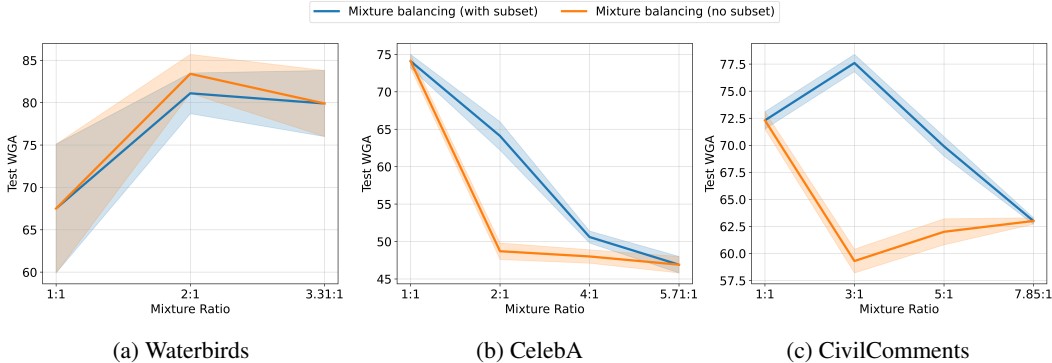

(a) Waterbirds          (b) CelebA          (c) CivilComments

Figure 7: **Mixture balancing ablation studies.** We perform two ablation studies on our mixture balancing method. First, we vary the class-imbalance ratio across the $x$ axis. On the left-hand side, using a class-imbalance ratio of 1:1 reduces to the subsetting technique; on the right-hand side, using the original class-imbalance ratio in the dataset reduces to upsampling. Second, we perform an ablation of whether subsetting is essential in mixture balancing. We plot our proposed method (which takes a subset of data based on the class-imbalance ratio, then performs upsampling) against the same method without subsetting, instead adjusting the class probabilities on the entire dataset as specified by the class-imbalance ratio. MultiNLI is class-balanced *a priori*, so we do not include it here.

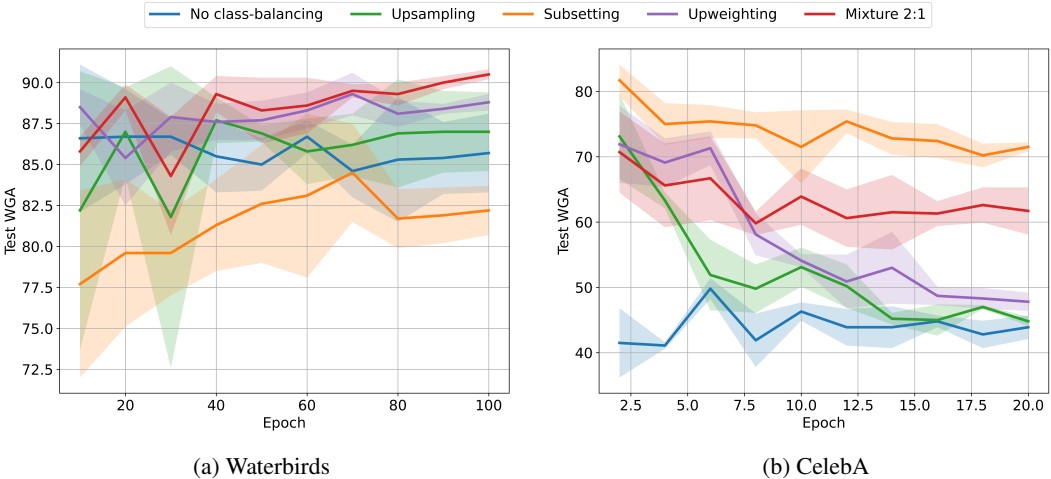

(a) Waterbirds          (b) CelebA

Figure 8: **Balancing behavior is consistent with Swin Transformer.** We demonstrate the effectiveness of our class-balanced *mixture method* when used in conjunction with a Swin Transformer (compare to the ConvNeXt-V2 results in Figure 2). Overall, we find our results are consistent across pretrained model families, with the model affecting the raw accuracies but typically not the relative performance of class-balancing techniques. We also corroborate the poor performance of subsetting on Waterbirds and the catastrophic collapse of upsampling and upweighting on Celeba from Figure 1.

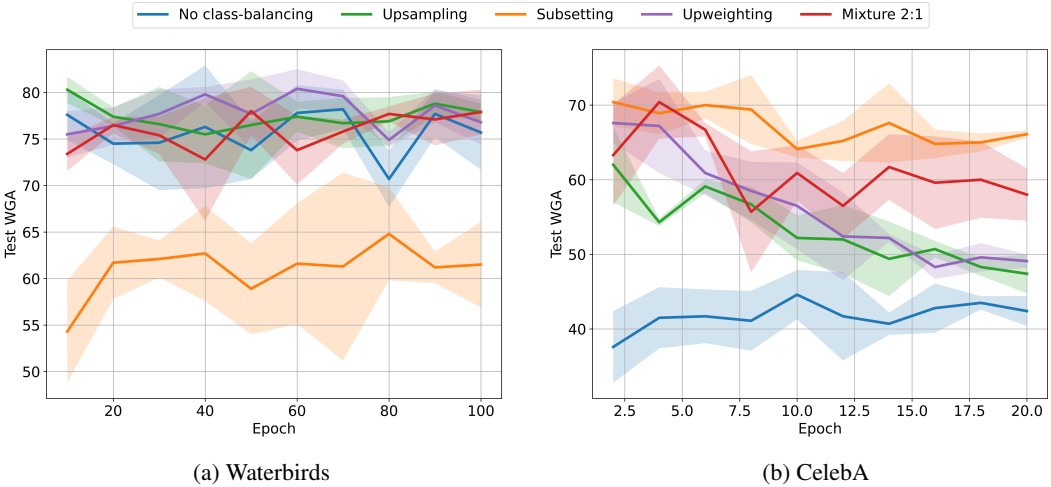

(a) Waterbirds

(b) CelebA

Figure 9: **Balancing behavior is consistent with ResNet model family.** We demonstrate the effectiveness of our class-balanced *mixture method* on another model family, ResNet (compare to the ConvNeXt-V2 results in Figure 2). Again, we find that our results are consistent and that the model architecture affects the raw accuracies but typically not the relative performance of class-balancing techniques. We also corroborate the poor performance of subsetting on Waterbirds and the catastrophic collapse of upsampling and upweighting on Celeba from Figure 1.

# C Additional Experiments for Section 4

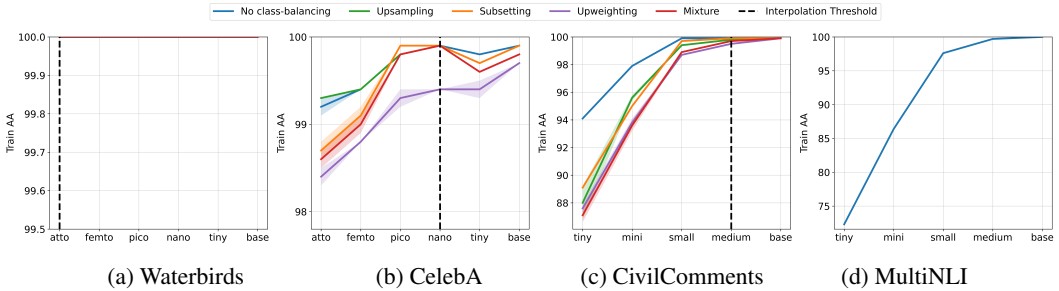

(a) Waterbirds     (b) CelebA     (c) CivilComments     (d) MultiNLI

Figure 10: **Average accuracy of scaled models.** We finetune each model size starting from pretrained checkpoints and plot the train average accuracy (AA) as well as the interpolation threshold, where *at least one seed* of the non-class-balanced model reaches $100\%$ training accuracy. (For example, CelebA does not interpolate with all three seeds). Average accuracy consistently increases with model size regardless of class-balancing, implying the scaling dynamics for AA and WGA are starkly different. Note that MultiNLI is class-balanced *a priori* and does not interpolate at any size.

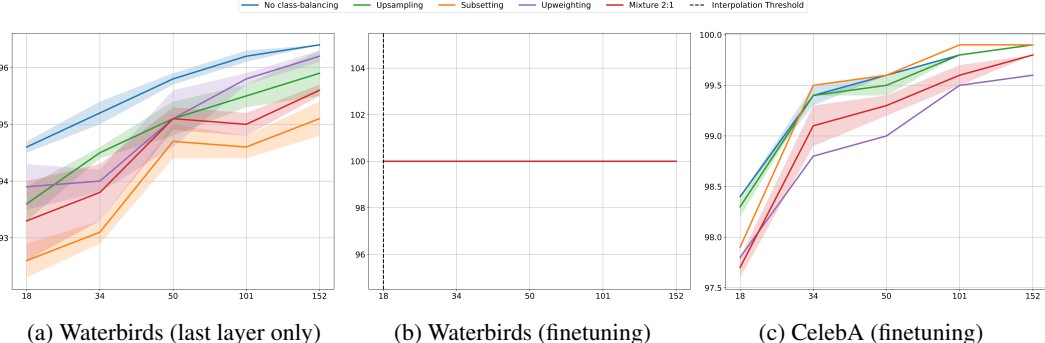

(a) Waterbirds (last layer only)     (b) Waterbirds (finetuning)     (c) CelebA (finetuning)

Figure 11: **Average accuracy of scaled ResNets.** We contrast the ResNet scaling behavior of [46] — who do not use class-balancing — to the scaling of class-balanced ResNets. We finetune each model size starting from pretrained checkpoints and plot the train average accuracy (AA) as well as the interpolation threshold, where the model reaches $100\%$ training accuracy. Similarly to Figure 10, average accuracy consistently increases with model size. We use SGD for last-layer training and AdamW for full finetuning.

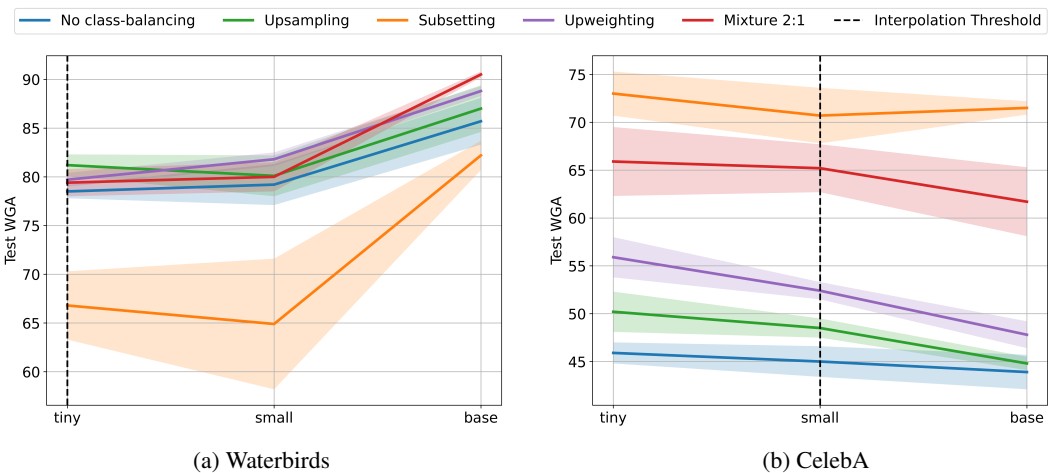

(a) Waterbirds  (b) CelebA

Figure 12: **Scaling behavior is consistent with Swin Transformer.** We exhibit the model scaling behaviour of a Swin Transformer, and compare it to that of a ConvNeXt-V2 (shown in Figure 3). We see that the scaling behaviour is consistent across pretrained model families, with the model affecting the raw accuracies but not the relative performance of class-balancing techniques.

# D   Additional Experiments for Section 5

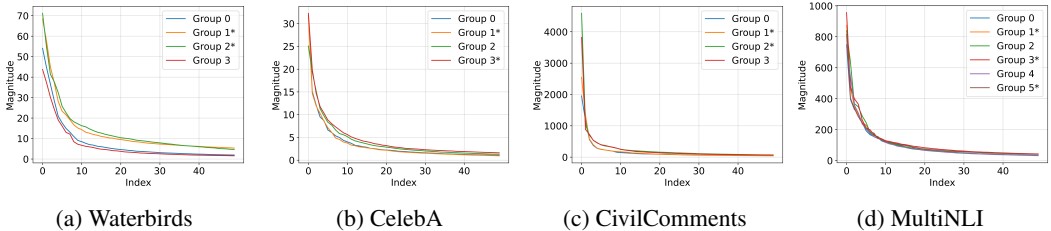

(a) Waterbirds      (b) CelebA      (c) CivilComments      (d) MultiNLI

Figure 13: **Additional eigenvalues of the group covariance matrices.** In contrast to Figure 5, we visualize the top 50 eigenvalues of the group covariance matrices for a ConvNeXt-V2 Nano finetuned on Waterbirds and CelebA and a BERT Small finetuned on CivilComments and MultiNLI. The models are finetuned using the best class-balancing method from Section 3 for each dataset. The group numbers are detailed in Table 2 and minority groups are marked with an asterisk. It becomes difficult to distinguish patterns between the groups in the lower eigenvalues, which is why we focus only on local properties of the top eigenvalues (*e.g.*, the spectral norm and the relative ordering of the groups). With that said, it would be interesting to explore power-law decay metrics [25], which characterize relatively global properties of the eigenspectrum, in future work.

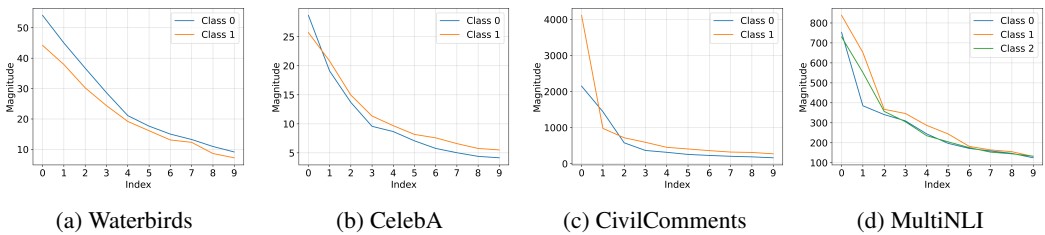

(a) Waterbirds      (b) CelebA      (c) CivilComments      (d) MultiNLI

Figure 14: **Class disparities are visible in the top eigenvalues of the class covariance matrices.** We visualize the mean, across 3 experimental trials, of the top 10 eigenvalues of the class covariance matrices for a ConvNeXt-V2 Nano finetuned on Waterbirds and CelebA and a BERT Small finetuned on CivilComments and MultiNLI. The standard deviations are omitted for clarity. The models are finetuned using the best class-balancing method from Section 3 for each dataset. The class numbers are detailed in Table 2. The minority class eigenvalues for CelebA and CivilComments are overall larger, while the reverse is true for Waterbirds, a slightly different conclusion than [25].

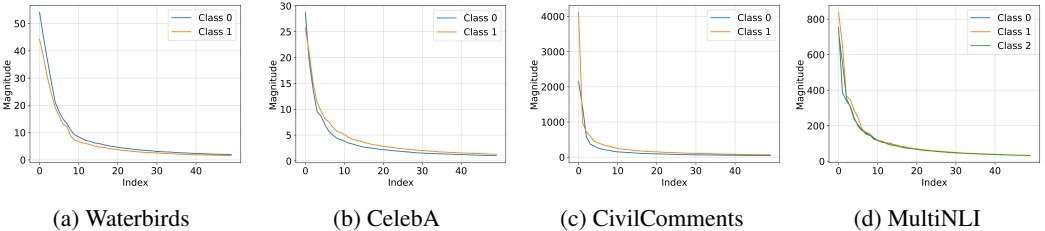

(a) Waterbirds      (b) CelebA      (c) CivilComments      (d) MultiNLI

Figure 15: **Additional eigenvalues of the class covariance matrices.** In contrast to Figure 14, we visualize the top 50 eigenvalues of the class covariance matrices for a ConvNeXt-V2 Nano finetuned on Waterbirds and CelebA and a BERT Small finetuned on CivilComments and MultiNLI. The models are finetuned using the best class-balancing method from Section 3 for each dataset. The class numbers are detailed in Table 2. Similar to the groups, it becomes difficult to distinguish patterns between the classes in the lower eigenvalues, which is why we again focus only on local properties of the top eigenvalues (*e.g.*, the spectral norm and the relative ordering of the classes).

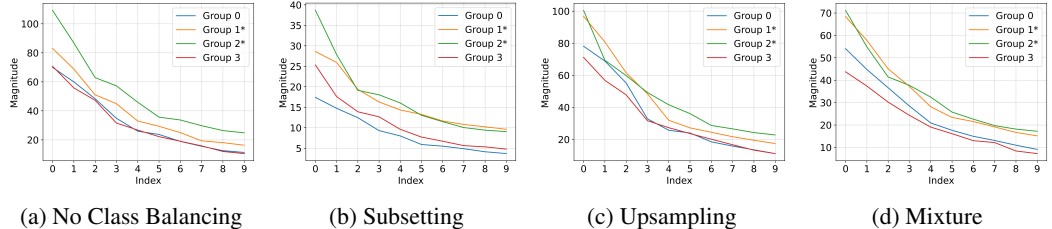

(a) No Class Balancing     (b) Subsetting     (c) Upsampling     (d) Mixture

Figure 16: **Group eigenvalue decay is consistent across balancing methods.** We visualize the mean, across 3 experimental trials, of the top 10 eigenvalues of the group covariance matrices for a ConvNeXt-V2 Nano finetuned on Waterbirds across all class-balancing methods. The standard deviations are omitted for clarity. Overall, we found that the magnitude of the eigenvalues is significantly affected by the chosen class-balancing method. However, the relative ordering of minority/majority group eigenvalues is consistent across class-balancing techniques. We note that the most drastic changes in the spectrum are induced by the subsetting method, which has the worst WGA by far for the Waterbirds dataset. These results suggest that optimal class-balancing may bring about additional stability in the representation.

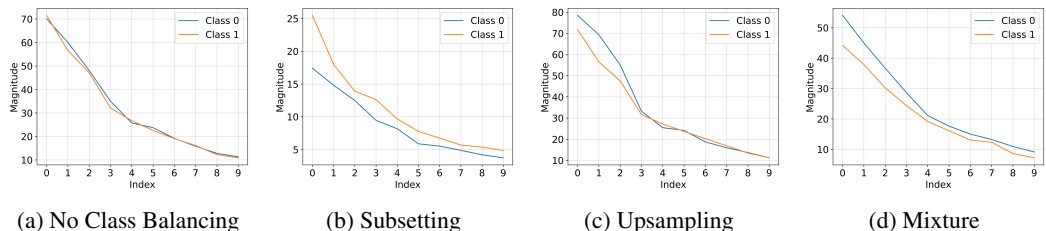

(a) No Class Balancing     (b) Subsetting     (c) Upsampling     (d) Mixture

Figure 17: **Class eigenvalue decay is consistent across balancing methods.** We visualize the mean, across 3 experimental trials, of the top 10 eigenvalues of the class covariance matrices for a ConvNeXt-V2 Nano finetuned on Waterbirds across all class-balancing methods. The standard deviations are omitted for clarity. Overall, we found that the magnitude of the eigenvalues is significantly affected by the chosen class-balancing method. However, the relative ordering of minority/majority group eigenvalues is consistent across class-balancing techniques. We note that the most drastic changes in the spectrum are induced by the subsetting method, which has the worst WGA by far for the Waterbirds dataset. These results suggest that optimal class-balancing may bring about additional stability in the representation.

Table 5: **Correspondence between $\rho^{(y)}$ and intra-class group accuracy disparity.** We compare $\rho^{(y)}$, the intra-class spectral norm ratio, to the difference in intra-class group accuracy. Each row represents a different experimental seed. Each cell contains a tuple with the class label for the class with largest value of $\rho^{(y)}$ paired with the class label for the class with the largest intra-class group test accuracy disparity, *i.e.*, $\mathrm{Acc}(g_{\mathrm{maj}}(y)) - \mathrm{Acc}(g_{\mathrm{min}}(y))$. We see that in most cases these classes correspond, suggesting an *explanatory power* of the spectral norm ratio. In particular, this correspondence is consistent throughout all trials of CelebA and CivilComments, the most class-imbalanced datasets we study.

| Seed | Waterbirds | CelebA | CivilComments | MultiNLI |
|---|---|---|---|---|
| 1 | (1, 1) | (1, 1) | (0, 0) | (0, 0) |
| 2 | (1, 1) | (1, 1) | (0, 0) | (0, 1) |
| 3 | (0, 1) | (1, 1) | (0, 0) | (2, 0) |

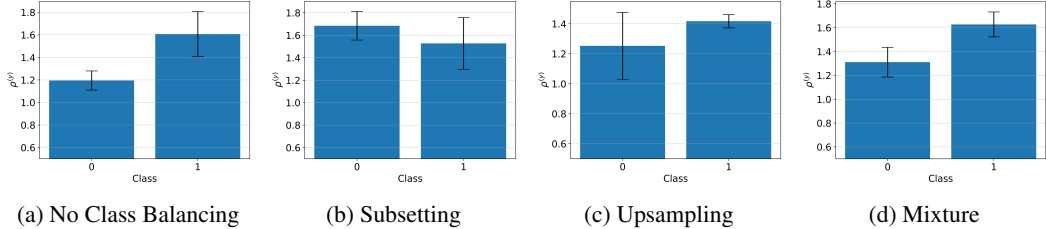

(a) No Class Balancing     (b) Subsetting     (c) Upsampling     (d) Mixture

Figure 18: **Spectral imbalance is consistent across balancing methods.** We plot the mean and standard deviation, across 3 experimental trials, of the intra-class spectral norm ratio $\rho^{(y)}$, or the ratio of the top eigenvalues of the minority and majority group covariance matrices, for each class $y \in \mathcal{Y}$. We compute this metric using a finetuned ConvNeXt-V2 Nano on Waterbirds. Overall, we found that the relative magnitudes of $\rho^{(y)}$ are consistent across class-balancing methods. We note that the most drastic change in the relative magnitudes of $\rho^{(y)}$ is induced by the subsetting method, which has the worst WGA by far for the Waterbirds dataset. These results suggest that optimal class-balancing may bring about additional stability in the representation.

# E Broader impacts, limitations, and compute

**Broader impacts.** We hope our work contributes to the safe and equitable application of machine learning and motivates further research in ML fairness. With that said, a potential negative outcome may arise if practitioners simply apply our techniques in place of conducting rigorous bias studies. Indeed, while our methods show improved fairness with respect to the worst-group accuracy metric, it is necessary to perform comprehensive evaluations with respect to multiple additional fairness criteria prior to model deployment.

**Limitations.** Our methods take advantage of the structure of spurious correlations; our insights would likely not transfer over to datasets which exhibit a more extreme *complete correlation* (*i.e.,* contain zero minority group data) [43, 30] or to more generic out-of-distribution generalization settings. A limitation of our mixture balancing method is that to achieve optimal performance, it requires a validation set with group annotations for selection of the best class-imbalance ratio [50, 31, 23, 26]. With that said, we show in Table 1 that worst-class accuracy [69] and the bias-unsupervised validation score of [60] are sufficient for model selection in the benchmarks we study.

**Compute.** Our experiments were conducted on two Google Cloud Platform (GCP) 16GB Nvidia Tesla P100 GPUs and two local 24GB Nvidia RTX A5000 GPUs. The spectral imbalance experiments in Section 5 were conducted on a GCP system with a 16-core CPU and 128GB of RAM. We believe our work could be reproduced for under $5000 in GCP compute credits, with a majority of that compute going towards running experiments over multiple random seeds.

