# OpenReview forum: "The Group Robustness is in the Details: Revisiting Finetuning under Spurious Correlations"
_NeurIPS.cc/2024/Conference — NeurIPS 2024 poster_

### Official Review · Reviewer_ax3f · 2024-06-20

**Soundness:** 3
**Presentation:** 3
**Contribution:** 3
**Rating:** 7
**Confidence:** 4

**Summary:**

The subject of the work is over-reliance on spurious correlations, which can often lead to poor performance on minority groups. The authors identify two failure modes of common class-balancing techniques: (1) class-balanced mini-batch finetuning experiences catastrophic collapse with standard hyperparameters on benchmark datasets, and (2) finetuning on a balanced subset of data can harm WGA when a small minority group is present in the majority class.

**Strengths:**

1. The authors provide important incremental advances to our understanding of subsetting and upsampling in this subfield; I.E., subsetting can be effective except when there is a small minority group present in the majority class, and ERM with class-balanced upsampling experiences a catastrophic collapse in test accuracy over the course of training.
2. The decision to study the impact of model scaling on worst-group accuracy in a new setting — finetuning pretrained models — which more closely resembles practical use-cases -- is a welcome incremental contribution.
3. The finding documented in Fig. 5 that the largest eigenvalue in each dataset belongs to a minority group and minority group eigenvalues are overall larger than majority group eigenvalues within the same class is a useful, if not terribly surprising, extension of [4].

**Weaknesses:**

1. In 3.2, the authors overclaim the strengths of their proposed method and their investigation of prior work in this section is spotty. The authors mention [1] as prior work; however, Deep Feature Reweighting, the main method explored in [1], requires group labels, so it is unclear what aspect of this work they are referencing. The authors do not address Automatic Feature Reweighting [2] in this section, which resolves spurious correlations using a weighted loss on a held-out dataset drawn simply from the training distribution. [2] retrains the last layer only, with weights prioritizing datapoints on which the base model performs poorly. The authors cannot and do not claim superiority without comparisons to existing work; however, it would still be extremely informative to see how their mixture balancing method compares to that of [2].
2. It would be helpful to have a section describing the motivation for why the authors chose to focus on subsetting and oversampling so heavily in the first section of the paper, included a section on spectral imbalance, but did not deeply investigate [3], which is also cited.

[1] Last Layer Re-Training is Sufficient for Robustness to Spurious Correlations

[2] Simple and fast group robustness by automatic feature reweighting

[3] Spuriosity Rankings: Sorting Data to Measure and Mitigate Biases

[4] Balanced Data, Imbalanced Spectra: Unveiling Class Disparities with Spectral Imbalance

**Questions:**

* Why was [1] referenced in Sec 3.2?
* Why was [2] not referenced in Sec. 3.2?
* What motivates the authors' choice to focus on subsetting and oversampling in Secs 3 and 4, and on spectral imbalance in 5, rather than a more general investigation of their proposed method's efficacy or a simple analysis of subsetting and oversampling in group robustness?

SUMMARY

As the paper stands, I think it is a weak accept; all of the contributions seem to be well supported by the evidence, but none of them seem likely to have an overwhelming impact. I will consider raising my score, however, if the weaknesses I mention are addressed.

**Limitations:**

The authors have adequately addressed the limitations and potential negative societal impact of their work.

---

> ### Author Rebuttal · Authors · 2024-08-06
>
> We graciously thank Reviewer ax3f for their detailed comments, questions, and references. We appreciate that the reviewer recognizes our contributions to the understanding of class-balancing methods, robustness impact of scaling pretrained models, and spectral analysis of model representations. Below, we provide responses to each of the reviewer’s points, combining weaknesses and questions as appropriate.
>
> ## Weakness 1, Question 1, Question 2
>
> We thank the reviewer for the insightful comments. Our intention was to avoid a comprehensive discussion of last-layer retraining methods, as we exclusively focus on model finetuning without held-out data or group labels. Our results are not directly comparable to those of DFR [1] and AFR [2] as they train on held-out data (up to 40K additional points) not included in standard model finetuning, as well as group labels (for training in [1] and model selection in [2] -- see the global response for mixture balancing model selection without group labels). On the other hand, since models finetuned with class-balancing are used as initializations for last-layer retraining [3], we expect our deeper understanding of finetuning phenomena to influence more sophisticated robustness algorithms.
>
> Our key reason for discussion of DFR [1] in Section 3.2 is not to compare to last-layer retraining, but instead to compare our mixture balancing method to their technique of averaging the last layer weights over multiple independently-sampled class-balanced data subsets, which like our proposed method intends to increase exposure to majority group samples without over-sampling the minority group. To our knowledge, this is the only technique in the literature which attempts to balance this trade-off. We did not cite AFR [2] in Section 3.2 because they do not mention this averaging technique, instead using a method akin to upweighting, and therefore make no attempt to increase exposure to majority class samples without over-sampling the minority class. Their approach of upweighting points on which the model does poorly is closely related to Just Train Twice (JTT) [4] and orthogonal to our methods. We will clarify these points in the paper.
>
> With that said, we agree with the reviewer that a more comprehensive discussion of where our mixture balancing method lies within the broader context of group robustness methods is appropriate. Please see the global response for further discussion.
>
> ## Weakness 2, Question 3
>
> We thank the reviewer for the detailed questions. While we considered including a simple theoretical analysis of subsetting and upsampling in class-balancing, a preliminary investigation revealed some interesting connections to the literature which we believe merit their own, purely theoretical submission in the future. In particular, we observed that while upsampling and upweighting have similar WGA dynamics, both differ greatly from subsetting, in contrast to [5] which proved a broader theoretical equivalence between the three techniques in the population setting. We believe theoretically investigating this discrepancy is an interesting direction, but it requires the development of some new techniques to analyze WGA dynamics when under-represented data is seen repeatedly by the model during training, which would be out of scope for this paper. We will add further technical discussion in our updated version.
>
> For the reviewer’s request for further investigation of class-balancing methods, we investigated an additional balancing technique (upweighting), proposed model selection without group labels, and performed additional experiments/ablations across two additional model families. Please see the global response for results and analysis.
>
> For the reviewer’s request of connecting spectral analysis to class-balancing, we re-ran our spectral analysis for Waterbirds with all balancing methods from the paper. Overall, we found that the magnitude of the eigenvalues is significantly affected by the chosen class-balancing method. However, the relative ordering of minority/majority group eigenvalues is consistent across class-balancing techniques. We note that the most drastic changes in the spectrum are induced by the subsetting method, which has the worst WGA by far for the Waterbirds dataset. These results suggest that optimal class-balancing may bring about additional stability in the representation. Please see the global response PDF for figures.
>
> Finally, we did not discuss Spuriosity Rankings [6] because it is focused on explainability/interpretability methods for discovering spurious features, which is orthogonal to our contributions. If the reviewer has a specific comparison they would like us to make with [6], we would be happy to address their concerns during the discussion phase.

---

> ### Comment · Reviewer_ax3f · 2024-08-08
>
> I would like to thank the authors for their detailed response. To acknowledge that some concerns have been addressed, I will update my score.

---

### Official Review · Reviewer_kRBT · 2024-07-12

**Soundness:** 2
**Presentation:** 3
**Contribution:** 3
**Rating:** 5
**Confidence:** 2

**Summary:**

The authors propose a class balancing scheme that both discards samples from the majority classes and upsamples majority classes as a way to improve worst group accuracy/robustness.

**Strengths:**

The authors propose a simple yet effective method that provides good performance for all of the datasets tested. The over-parametrization and spectral imbalance empirical analysis are of independent interest.

**Weaknesses:**

Limited empirical Evidence: Although the benchmark datasets are taken from prior work (Idrissi et. al)  and indeed popular,  most of the claims are substantiated by experiments on only three settings, in which the two baseline methods show a different behaviour. The claim that their method works better on a wider range of settings (without hyperparameter tuning) could be supported by more empirical evidence.

Relevance of the Feature Spectral Analysis: Although it is interesting and informative/insightful, there is no discussion on how the spectral features change between the proposed method and baseline  methods (subsetting/upsampling) - i.e. , how the balanced is achieved. Thus it seems unrelated to the discussion about the proposed method.

Ratio ablation and choice: There is not much discussion about the choice of mixture ratio  (again this is important because there is emphasis in this method performing well *without* hyperparameter tuning). The ratio ablation included in the supplementary (figure 7) only includes two values per dataset (apart from the baselines). The two values shown do have similar performance so if this holds with more points, this could better illustrate the robustness of the method w.r.t this hyperparameter.

**Questions:**

Did you tune the mixture ratio? Why don't you compare to other class balancing techniques apart from subsetting and oversampling?
Have you tried other datasets or other group-imbalance settings?

**Limitations:**

Yes

---

> ### Author Rebuttal · Authors · 2024-08-06
>
> We warmly thank Reviewer kRBT for their detailed comments and questions. We appreciate that the reviewer recognizes the improved performance of our mixture balancing method and the independent interest of our model scaling and spectral analysis experiments. Below, we provide responses to each of the reviewer’s points, combining weaknesses and questions as appropriate.
>
> ## Weakness 1, Question 3
>
> We appreciate the reviewer’s request for further empirical evidence and evaluation settings. In the global response, we investigate an additional balancing technique (upweighting), propose model selection without group labels, and perform additional experiments/ablations across two additional model families. Please see the global response for results and analysis.
>
> ## Weakness 2
>
> For the reviewer’s request of connecting spectral analysis to class-balancing, we re-ran our spectral analysis for Waterbirds with all balancing methods from the paper. Overall, we found that the magnitude of the eigenvalues is significantly affected by the chosen class-balancing method. However, the relative ordering of minority/majority group eigenvalues is consistent across class-balancing techniques.
>
> We note that the most drastic changes in the spectrum are induced by the subsetting method, which has the worst WGA by far for the Waterbirds dataset. These results suggest that optimal class-balancing may bring about additional stability in the representation. Please see the global response PDF for figures.
>
> ## Weakness 3, Question 1
>
> We agree with the reviewer that while performing model selection with respect to worst-group accuracy is a common assumption in the literature [1, 4, 8, 9], it is nevertheless unrealistic when group labels are not available. In the global response, we address these concerns by investigating several methods for model selection without group labels -- using 3-4 mixture ratios per dataset -- and conclude that our methods are robust to validation without group labels (indeed, validation with respect to worst-class accuracy is sufficient). Please see the global response for results and analysis.
>
> ## Question 2
>
> We thank the reviewer for the question and we address it in the global response. In addition to subsetting, upsampling, and mixture balancing, we have investigated another common class-balancing technique called *upweighting* not studied by [7]. In this method, minority class samples are directly upweighted in the loss function by the class-imbalance ratio, and it is used by state-of-the-art group robustness algorithms including AFR [2]. We find that upweighting experiences a similar catastrophic collapse as upsampling, even though they are only equivalent *on average* over the upsampling probabilities (i.e., not in practice). Please see the global response PDF for figures.

---

### Official Review · Reviewer_E5Mf · 2024-07-12

**Soundness:** 2
**Presentation:** 3
**Contribution:** 2
**Rating:** 4
**Confidence:** 4

**Summary:**

The paper tries to study the fundamental properties of fine-tuning DNNs and worst group accuracy in the presence of spurious correlations. The effort focuses on revealing nuances that were not clear. They conducted experiments with both 2 vision and 2 NLP datasets with spurious correlations and subgroup labels present. Within the authors' setup, the experiments discuss the (1) impacts of class-balancing on group robustness (2) proposed approach to mitigate (3) model scaling's impact and (4) spectral imbalance's effect.

**Strengths:**

1. The paper conduct experiments to challenge the belief of (1) "overparameterization helps or hurts robustness" and (2) show the impact of different class-balancing methods,

2. They conducted comprehensive experiments to explore and reveal the "nuances" in different directions across vision & language tasks, and propose suggestions when facing those scenarios.

3. They propose a mixture method to outperform 2 prior common practices, subsetting and mini-batch class-balancing upsampling. The central idea of the mixture method is interpolate subsetting and upsampling.

**Weaknesses:**

1. This paper aim to study the "fundamental properties of fine-tuning" but focused on ConvNext only and didn't experiment with another popular pre-trained model family such as ViTs.

2. The paper keeps reiterating the "nuanced", however, I believe the those nuances are general prior beliefs, I would hope the authors can provide some ablations to clarify those nuances and potentially provide clear guidelines, since all methods have nuances.

3. The findings of this paper have some dependencies, the success cases seem to depend on a specific or "appropriate" setup to work.

4. The datasets the authors use has group labels in training set, the authors should at least study the how maybe other training paradigms such as GroupDRO, etc., fare in the authors setting and proposed method, otherwise I believe the scope would be relatively narrow.

5. The datasets used are binary or trinary. In a more realistic setting multi-class setting, the nuances should be discussed.

**Questions:**

1. Authors have put summaries at the end of each experiments. However, do the authors have a clearer guidelines for what practitioners should do under each "nuanced" scenarios, or just suggest there are more "nuanced" scenarios?

2. Datasets authors used are all binary classifications, except MultiNLI (trinary), what would the nuances look like in a realistic multi-class setting? Would it create more nuances? [1] has an not ideal but acceptable multi-class dataset with group labels.

3. What would the nuances look like in a ViT or its variants setting? As ViTs are know to require significant pretraining and downstream finetuning.

---------------------------------------
[1] Spawrious [Lynch, et. al, 2023]

**Limitations:**

Yes.

---

> ### Author Rebuttal · Authors · 2024-08-06
>
> We graciously thank Reviewer E5Mf for their insightful comments and questions. We appreciate that the reviewer recognizes our comprehensive experiments which challenge existing notions in the literature and leverage previously unknown nuances to improve model performance. Below, we provide responses to each of the reviewer’s points, combining weaknesses and questions as appropriate.
>
> ## Weakness 1, Question 3
>
> We appreciate the suggestion to replicate our experiments with other popular pretrained model families. We have implemented these experiments in two settings: Swin Transformer [10], a more efficient variant of Vision Transformer [11] with three pretrained sizes available, and ResNet [12], the most popular convolutional backbone with five pretrained sizes available. Overall, we find our results are consistent across pretrained model families, with the model affecting the raw accuracies but typically not the relative performance of class-balancing techniques or scaling behavior. Please see the global response PDF for figures.
>
> ## Weakness 2, Question 1
>
> We appreciate the reviewer’s comments and we agree that we could be more explicit about our recommendations for practitioners. Below, we provide more practical recommendations based on the “nuances” we propose in our paper.
>
> 1. Since group proportions are unknown in practice, a good proxy for which class-balancing technique to use is the size of one’s dataset. If the dataset is on the smaller side (a couple thousand points, like Waterbirds), one should use upsampling or mixture balancing. If the dataset size is larger than 100K (like CivilComments), subsetting or mixture balancing is recommended.
>
> 2. With appropriate pretraining and class-balancing, model scaling is generally beneficial for group robustness. Therefore, practitioners should utilize the largest model appropriate for their use case (especially if it does not interpolate, like MultiNLI).
>
> 3. Practitioners should typically apply a post-hoc group robustness technique like AFR [2] or SELF [3] after finetuning, since as seen in the global response, class-balancing can only push WGA so far. With that said, the performance of the initial finetuned model affects downstream WGA (especially due to representation quality for last-layer methods [8]), so it is still worthwhile to carefully select model pretraining parameters.
>
> We would appreciate it if you could point out the general prior beliefs you refer to. The literature appears to reflect a lack of consensus even with respect to phenomena as seemingly simple as model scaling. For example, in our related work section, we discuss conclusions from the literature which state that model scaling could either be beneficial or detrimental to robustness performance, depending on the particular setting and assumptions used for analysis.
>
> In the global response, we address the reviewer’s request for additional experiments/ablations by investigating an additional balancing technique (upweighting), proposing model selection without group labels, and performing additional experiments/ablations across two additional model families. If there is a specific ablation the reviewer wishes to see that has not been addressed in the global response, please let us know and we would be happy to address it during the discussion phase.
>
> ## Weakness 4
>
> We agree with the reviewer that a more comprehensive discussion of where our mixture balancing method lies within the broader context of group robustness methods is appropriate. We would like to clarify that, in contrast to Group DRO [9], our methods **do not** use group labels in the training dataset (they are discarded before training). Please see the global response for a more explicit comparison with Group DRO and other methods, where we clearly delineate if held-out data or group labels are utilized.
>
> Furthermore, while performing model selection with respect to worst-group accuracy is a common assumption in the literature [1, 4, 8, 9], it is nevertheless unrealistic when group labels are not available. We investigate several methods for model selection and conclude that our methods are robust to validation that does not use group labels (indeed, validation with respect to worst-class accuracy is sufficient). Please see the global response for results and analysis.
>
> ## Weakness 5, Question 2
>
> We appreciate the reviewer’s suggestion and we agree that it would be interesting to observe the behavior of our methods on multi-class datasets. However, the dataset Spawrious [15] suggested by the reviewer is class-balanced *a priori* and hence not suitable for evaluating class-balancing methods (similarly to MultiNLI). With that said, we were able to run model scaling experiments. We used the most rigorous O2O Hard split and include the results below:
>
> | ConvNeXt-V2 Version | WGA (3 seeds) |
> |---------------------|---------------|
> | Atto (3.4M params)  | 24.8 +/- 2.0  |
> | Femto (4.8M params) | 17.3 +/- 1.4  |
> | Pico (8.6M params)  | 44.3 +/- 2.6  |
> | Nano (15.0M params) | 38.8 +/- 2.7  |
> | Tiny (27.9M params) | 37.3 +/- 3.4  |
> | Base (87.7M params) | 57.1 +/- 4.0  |
>
> Despite the trend being less monotone than other datasets, it is clear that scaling pretrained and class-balanced models is broadly beneficial for robustness on Spawrious, especially as one nears the 100M parameters mark.
>
> We anticipate that our class-balancing methods would generalize to multi-class datasets without additional nuance, since the failure modes we identify for each method remain the same no matter the number of classes. Specifically: (1) for upsampling, any small class will be over-represented during long training runs no matter the number of classes, so we expect overfitting to the minority group within this class and poor WGA as a result; and (2) for subsetting, any minority group within a large class will be severely downsampled no matter the number of classes, so subsetting will disproportionately harm that group.

---

> > ### Comment · Reviewer_E5Mf · 2024-08-13
> >
> > I would like to thank the authors for the response and acknowledge that some concerns have been addressed.

---

### Official Review · Reviewer_Rwfv · 2024-07-13

**Soundness:** 1
**Presentation:** 1
**Contribution:** 1
**Rating:** 2
**Confidence:** 5

**Summary:**

This paper experimentally analyzes the impact of class balancing on group robustness.

**Strengths:**

This research contributes to the community by investigating spurious correlations in machine learning models' reliance.

**Weaknesses:**

- Due to significant variations in experimental results across datasets, generalization based on these results appears challenging. The case-by-case description of experimental outcomes seems more akin to a technical report rather than a conference paper.

- The format of the citations is incorrect. The incorrect basic formatting makes the paper appear unprofessional.

- The Related work subsection should be a section.

- Lines 112-117 are not essential for the main content of the paper.

**Questions:**

See the weaknesses

**Limitations:**

Yes, they have addressed.

---

> ### Author Rebuttal · Authors · 2024-08-06
>
> We warmly thank Reviewer Rwfv for their comments and suggestions. Below, we provide responses to each of the reviewer’s points, combining weaknesses and questions as appropriate.
>
> ## Weakness 1
>
> We thank the reviewer for the relevant comments. While robustness performance indeed differs across datasets, our explicit goal is to elucidate these nuances, analyze their causes and effects, and provide the community with a rigorous foundation for further research. Furthermore, we do find a potentially generalizable pattern for our class-balancing results. Specifically, we investigate two new failure modes of class-balancing which depend on the *particular group/class structure of each dataset*: (1) mini-batch upsampling and loss upweighting experience catastrophic collapse with standard hyperparameters when imbalance is large, and (2) removing data to create a class-balanced subset can harm WGA when a small minority group is present in the majority class.
>
> We further investigate the generalizability of our results in the global response. In particular, our observations remain consistent even when tested on new class-balancing methods, without group labels for model selection, and in new settings including two different model families.
>
> Finally, we remark that case-by-case descriptions of experimental outcomes have strong precedent in the community. For example, Izmailov et al. [8] investigate the case-by-case effects of architecture, pretraining dataset, and regularization on multiple benchmark datasets. Similarly to us, they analyze how their results vary across different datasets and leverage their insights to improve model performance and make practical recommendations.
>
> ## Weakness 2
>
> We believe the reviewer is referencing the fact that we utilized alphabetical/lastname citations instead of numerical citations. We respectfully refer the reviewer to line 150 in the NeurIPS 2024 Formatting Instructions, which states “Any choice of citation style is acceptable as long as you are consistent.” If the reviewer was referencing some other aspect of our citations, please feel free to clarify the specific concerns, and we would be happy to address them during the discussion phase.
>
> ## Weakness 3, Weakness 4
>
> We thank the reviewer for the suggestions and we will include them in our updated version.

---

> ### Comment · Reviewer_Rwfv · 2024-08-08
>
> Thank you for your detailed response. I would like to further elaborate on my concerns.
>
> Q1. **Inconsistencies in Claims**: There appears to be a contradiction within the manuscript regarding the effectiveness of subsetting in the presence of a small minority group within the majority class. Specifically, in Lines 157-159, the authors state that subsetting improves WGA when there is a small minority group within the majority class. However, in Figure 1, the results for the Waterbirds dataset show that subsetting performs worse than no class-balancing. Additionally, in Lines 188-189, the authors mention that "subsetting can be effective except when there is a small minority group present in the majority class," which directly contradicts the earlier statement. Could you clarify which of these claims is correct?
>
> Q2. **Lack of Controlled Variables**: The experimental setup seems to lack control over various factors, making it difficult to draw logical or theoretical conclusions from the results. For instance, regarding the claim made in Question 1, is it the case that the authors derived the conclusion from the results shown in Figure 1? In that case, there is a concern that the proposed reason for the difference in performance observed in the Waterbirds dataset compared to CelebA and CivilComments might not be a generalizable finding but rather a result of simply choosing one dataset out of several with inherent differences. For instance, unlike CelebA and CivilComments, the Waterbirds dataset shows lower performance for subsetting, which could be attributed to the significantly smaller training set size (Waterbirds: 4,795 samples; CelebA: 162,770 samples; CivilComments: 269,038 samples). The smaller size of the Waterbirds dataset could lead to a substantial decrease in overall performance when further reduced by subsetting, which also can explain the observed results. Additionally, the degree of group imbalance within classes is not as significant in CelebA and CivilComments, which distinguishes them from the Waterbirds dataset and could be a factor influencing WGA. These factors, if uncontrolled, could have influenced the results, and I wonder if they were considered in the analysis. How can we distinguish between the interpretation of the paper and these alternative explanations? Considering this, I find it challenging to fully agree with the statement, which underpins the paper, that "removing data to create a class-balanced subset can harm WGA when a small minority group is present in the majority class.
>
> Q3. **Dataset Suitability and Analysis**: Given that the Waterbirds dataset shows high WGA for no class-balancing methods and the minimal difference compared to upsampling, could it be that this dataset is less sensitive to the degree of imbalance in the training set and therefore not well-suited for analyzing the behavior of class-balancing methods? If Waterbirds is excluded, it seems that conducting case studies on class balancing methods with only the CelebA and CivilComments datasets might explore only a limited range of cases.
>
> Q4. When comparing the performance of upsampling, evaluating it by epoch might not provide a fair comparison, as upsampling leads to more iterations. Wouldn't it be more rigorous to compare the results based on the number of iterations with the same batch size?
>
> Q5. (minor question) I understand why ConvNext-V2 was chosen for the experiments in Section 4, but wouldn't it be beneficial for the field to also include results using ResNet50 for the other experiments? Additionally, I’m curious whether the results in Figures 1 and 2 would be consistent if ResNet50 were used.

---

> ### Author Response · Authors · 2024-08-11
>
> Thank you for your continued engagement. We provide responses to your questions below.
>
> ## Question 1
> We appreciate the catch; we agree that our sentence in lines 158-159 was unclear. We will rewrite the sentence to say “…it can in fact improve WGA conditional on the **lack of** a small minority group within the minority class.” Thus, the claim is consistent with Figure 1 and lines 188-189.
>
> ## Question 2
> We agree with the reviewer that a more controlled investigation would be beneficial. To show that our conclusions hold in a controlled environment, we extend the synthetic dataset of [Sagawa et al., 2020: An investigation of why overparameterization exacerbates spurious correlations] to our class-imbalanced setting.
>
> We generate a dataset of $100000$ points with labels $y\in \\{-1,1\\}$ and spurious attributes $s\in\\{-1,1\\}$ as follows. Each $(y, s)$ group has its own distribution over input features $x = [x_{core},x_{spu}]\in\mathbb{R}^{2d}$, corresponding to core features in $\mathbb{R}^d$ generated from the label $y$ and spurious features in $\mathbb{R}^d$ generated from the spurious attribute $s$:
> $$x_{core}\sim \mathcal{N}(y\vec{1}, \sigma^2_{core} \mathbf{I}\_d)\qquad x_{spu}\sim \mathcal{N}(s\vec{1}, \sigma^2_{spu} \mathbf{I}_d).$$
>
> We set $d=100$, $\sigma^2_{core}=100$, and $\sigma^2_{spu}=1$ following [Sagawa et al., 2020]. Different from their setup, we generate the data according to a class-imbalanced distribution where a small minority group is present within the majority class. To do so, we introduce a variable $\alpha\in [0.5, 1.0)$ which controls the size of the minority group within the majority class. Our dataset composition is detailed below:
>
> | Group (y, s) | Number of data |
> |--------------|----------------------|
> | (-1, -1)     | $45000 \alpha$       |
> | (-1, 1)      | $45000 (1 - \alpha)$ |
> | (1, -1)      | $5000$                 |
> | (1, 1)       | $5000$                 |
>
> We train a two-layer ReLU neural network with hidden dimension $64$ using full-batch gradient descent with learning rate $0.01$ and momentum $0.9$. We class-balance with the subsetting method and evaluate on a held-out dataset of $100000$ points balanced across groups. Our WGA results are detailed below, averaged over ten seeds:
>
> | $\alpha$ | Subsetting | No class-balancing |
> |----------|------------|--------------------|
> | 0.5      | 45.2       | 2.3                |
> | 0.6      | 37.6       | 1.1                |
> | 0.7      | 28.1       | 0.71               |
> | 0.8      | 23.1       | 0.35               |
> | 0.9      | 8.5        | 0.1                |
>
> As seen in the table, the smaller the size of the minority group in the majority class (i.e., the larger $\alpha$ is), the worse the subsetting method performance becomes. We believe this contributes to the justification of our conclusions in a controlled environment, and we would be happy to discuss further if the reviewer has additional concerns.
>
> ## Question 3
> We appreciate the reviewer’s concern. We believe the sensitivity to the degree of imbalance is not just a property of the dataset, but of the model as well. In Figure 3 in the global response PDF, we show that Swin Transformer Base exhibits a larger degree of sensitivity on Waterbirds compared to ConvNeXt-V2. Below, we include a table with a more explicit comparison (WGA averaged over 3 seeds):
> | Architecture          | Mixture balancing | Upweighting | Upsampling | No class-balancing | Subsetting |
> |-----------------------|-------------------|-------------|------------|--------------------|------------|
> | ConvNeXt-V2 Base      | 81.1              | 80.2        | 79.9       | 80.4               | 67.5       |
> | Swin Transformer Base | 90.5              | 88.8        | 87.0       | 85.7               | 82.2       |
>
> Contrary to the reviewer’s concern that “the Waterbirds dataset shows high WGA for no class-balancing methods and the minimal difference compared to upsampling,” the suboptimality of no-class balancing and upsampling on Waterbirds is clearly observed for Swin Transformer Base. Together with the strong precedent in the literature of evaluating class-balancing methods on Waterbirds [3, 7], we believe Waterbirds is well-suited for our investigation.
>
> ## Question 4
> We would like to clarify that upsampling **does not** lead to more iterations compared to training without class-balancing. In upsampling, we sample the mini-batches uniformly over the classes, but each mini-batch contains the same amount of data and we train for the same amount of steps as without class-balancing. We will make this more explicit in the paper.
>
> ## Question 5
> We agree that these results will be important for the community. In Figures 2 and 3 in the global response PDF, we replicated our class-balancing and scaling experiments for different class-balancing methods using the ResNet family (including ResNet50) and Swin Transformer. We will include more comprehensive ResNet50 results in the updated version of the paper.

---

> > ### Comment · Reviewer_Rwfv · 2024-08-14
> >
> > I appreciate the authors' efforts to address my concerns.
> >
> > Q2: My concern is that it's incorrect to draw conclusions as if it were a controlled environment when dealing with real datasets in an uncontrolled environment. While I'm grateful for the toy example provided, it seems difficult to assume that the results from the toy example would apply to real data.
> >
> > Q3. My concern is that the proposed findings lack generalizability across datasets or models. This issue cannot be adequately addressed by identifying a single case of suboptimality. Furthermore, while the authors claim to have confirmed the suboptimality of no-class balancing and upsampling on Waterbirds with the Swin Transformer Base, but there appears to be minimal performance difference between upsampling and no-class balancing, even when utilizing the Swin Transformer Base.
> >
> > Q5. Contrary to the authors' response, I could not find the ResNet50 results in the attached document. I only see results for ResNet18 and ResNet152 from the ResNet family.
> >
> > Therefore, I still have significant concerns about the lack of unified experimental settings in this paper, making fair comparisons difficult, and about drawing conclusions in an uncontrolled environment. Consequently, I will maintain my score.

---

### Author Rebuttal · Authors · 2024-08-06

We graciously thank all reviewers for their time and insights. Here, we provide new comparisons and experiments of interest to multiple reviewers.
## Additional class-balancing technique (all reviewers)
In addition to subsetting, upsampling, and mixture balancing, we investigated another common class-balancing technique called *upweighting* not studied by [7]. In this method, minority class samples are directly upweighted in the loss function by the class-imbalance ratio, and it is used by state-of-the-art group robustness algorithms including AFR [2]. We found that upweighting experiences a similar catastrophic collapse as upsampling, even though they are only equivalent *on average* over the upsampling probabilities (i.e., not in practice). Please see the attached PDF for figures.
## Model selection without group labels (all reviewers)
While performing model selection with respect to worst-group accuracy is a common assumption in the literature [1, 4, 8, 9], it is nevertheless unrealistic when group labels are not available. To address this, we re-ran mixture ratio tuning with respect to both worst-class accuracy [13] and the recently proposed *bias-unsupervised validation score* [14], which do not use any group labels for model selection. We performed model selection over at least 3-4 mixture ratios per dataset, and below we list the ratio which maximized each metric as well as its average WGA over 3 seeds.
| Validation Metric            | Group Labels | Waterbirds    | CelebA     | CivilComments |
|----------------------------|--------------|---------------|------------|---------------|
| Bias-unsupervised Score [14] | No           | 3.31:1 (79.9) | 1:1 (74.1) | 3:1 (77.6)    |
| Worst-class accuracy [13]    | No           | 2:1 (81.1)    | 1:1 (74.1) | 3:1 (77.6)    |
| Worst-group accuracy         | Yes          | 2:1 (81.1)    | 1:1 (74.1) | 3:1 (77.6)    |

Overall, both worst-class accuracy and the bias-unsupervised validation score performed well for our use case; in fact, similarly to the commonly used worst-group-accuracy score. All of the scores enable tuning of the mixture ratio without any group labels.
## Additional model families (all reviewers)
We appreciate the suggestions to replicate our experiments with other popular pretrained model families. We implemented these experiments in two settings: Swin Transformer [10], a more efficient variant of Vision Transformer [11] with three pretrained sizes available, and ResNet [12], which has five pretrained sizes available.  Overall, we find our results are consistent across pretrained model families, with the model affecting the raw accuracies but typically not the relative performance of class-balancing techniques or the impact of model scaling. Please see the attached PDF for figures.

## Connection of spectral analysis to class-balancing (kRBT, ax3f)
We thank the reviewers for their suggestion to compare spectral properties among different class-balancing techniques. We re-ran our spectral analysis for Waterbirds with all balancing methods from the paper. Overall, we found that the magnitude of the eigenvalues is significantly affected by the chosen class-balancing method. However, the relative ordering of minority/majority group eigenvalues is consistent across class-balancing techniques. Please see the attached PDF for figures.

We note that the most drastic changes in the spectrum are induced by the subsetting method, which has the worst WGA by far for the Waterbirds dataset. These results suggest that optimal class-balancing may bring about additional stability in the representation.

## Contextualization with broader group robustness methods (E5Mf, kRBT, ax3f)

We agree with the reviewers that a more comprehensive discussion of where our mixture balancing method lies within the broader context of group robustness methods is appropriate. We view mixture balancing as a soft upper bound on the performance of the simple baseline of ERM with class-balancing alone, as we select the optimal interpolation between subsetting and upsampling. We naturally expect this simple baseline to still perform worse than more sophisticated (and often more computationally intensive) group robustness methods. We compare the WGA of referenced methods below:

| Method                   | Held-out Data | Group Labels | Waterbirds | CelebA | CivilComments |
|--------------------------|---------------|--------------|------------|--------|---------------|
| DFR [1]                  | Yes           | Yes (train)  | 91.1       | 89.4   | 78.8          |
| AFR [2]                  | Yes           | Yes (val)    | 90.4       | 82.0   | ---           |
| Group DRO-ES [8, 9]      | No            | Yes (train)  | 90.7       | 90.6   | 80.4          |
| Just Train Twice [4, 7]  | No            | No           | 85.6       | 75.6   | ---           |
| Mixture balancing (ours) | No            | No           | 81.1       | 74.1   | 77.6          |
| ERM (no balancing)       | No            | No           | 79.4       | 48.3   | 58.7          |

Keeping in mind that the goal of mixture balancing is **not** to achieve state-of-the-art group robustness performance, we can see that the addition of held-out data and group labels improves WGA significantly over JTT and mixture balancing on Waterbirds and CelebA. However, on CivilComments, mixture balancing is competitive with state-of-the-art robustness methods, corroborating the conclusion of [3] that *class-balancing is sufficient for group robustness* in some cases.

In terms of guidelines for practitioners: our results show they should typically apply a post-hoc group robustness technique like AFR [2] or SELF [3] after finetuning, since class-balancing can only push WGA so far. With that said, the performance of the initial finetuned model affects downstream WGA (especially due to representation quality for last-layer methods [8]), so it is still worthwhile to carefully understand model pretraining parameters.

---

### Author Response · Authors · 2024-08-06
**References for Author Rebuttal**

We include our list of references for the author rebuttal here, as we did not have space in the individual responses. The numbering of references is consistent across all reviewer responses.

**[1]** Kirichenko et al. “Last Layer Re-training is Sufficient for Robustness to Spurious Correlations.” ICLR 2023.

**[2]** Qiu et al. “Simple and fast group robustness by automatic feature reweighting.” ICML 2023.

**[3]** LaBonte et al. “Towards Last-layer Retraining for Group Robustness with Fewer Annotations.” NeurIPS 2023.

**[4]** Liu et al. “Just train twice: Improving group robustness without training group information.” ICML 2021.

**[5]** Stromberg et al. “Robustness to Subpopulation Shift with Domain Label Noise via Regularized Annotation of Domains.” ArXiv 2024.

**[6]** Moayeri et al. “Spuriosity Rankings: Sorting Data to Measure and Mitigate Biases.” NeurIPS 2023.

**[7]** Idrissi et al. “Simple data balancing achieves competitive worst-group-accuracy.” CLeaR 2022.

**[8]** Izmailov et al. “On feature learning in the presence of spurious correlations.” NeurIPS 2022.

**[9]** Sagawa et al. “Distributionally robust neural networks for group shifts: On the importance of regularization for worst-case generalization.” ICLR 2020.

**[10]** Liu et al. “Swin Transformer: Hierarchical Vision Transformer using Shifted Windows.” ICCV 2021.

**[11]** Dosovitskiy et al. “An Image is Worth 16x16 Words: Transformers for Image Recognition at Scale.” ICLR 2021.

**[12]** He et al. “Deep Residual Learning for Image Recognition.” CVPR 2016.

**[13]** Yang et al. “Change is Hard: A Closer Look at Subpopulation Shift.” ICML 2023.

**[14]** Tsirigotis et al. “Group Robust Classification Without Any Group Information.” NeurIPS 2023.

**[15]** Lynch et al. “Spawrious: A Benchmark for Fine Control of Spurious Correlation Biases.” ArXiv 2023.

---

### Decision · Program_Chairs · 2024-09-25

**Decision:**

Accept (poster)

**Comment:**

This paper is concerned with spurious correlations, which are closely related to the issue of poor performance on minority data slices or groups. Typical solutions involve re-balancing through a variety of means, but this can accidentally cause issues in certain settings (but not in others). The authors dig into these issues via carefully-crafted experiments, leading to a solid set of insights. As an example, the authors describe behavior from previous published work that was mysterious: using a popular previous method decreases the worst group accuracy during training; this is explained by overfitting on repeatedly-sampled minority points. The authors also provided new approaches inspired by these observations; these are simple but powerful (for example, the mixture approach).

Reviews for this paper were somewhat mixed. One reviewer was quite negative; however, I felt that the authors were able to respond to all of the concerns raised. Others were very positive. I tended to agree with the latter—the paper has a nice contribution overall. Papers that dig into critical details and explain difficult-to-understand behaviors are very valuable, and this paper does that in spades.